# Thyrotropin-releasing hormone neurons of different hypothalamic nuclei increase energy expenditure

Andreea Constantinescu [1,7], Akila Chandrasekar[1,7], Lena Kleindienst[1], Luca Höhne[1], Natalia Da Silva Lima [2], Marius Richter[1], Vanessa Neve[1], Frauke Spiecker[1], Ines Stölting[1], Wiebke Brandt[1], Urte Matschl[3], Jan Wenzel [1], Rebecca Oelkrug[4], Vincent Prevot [5], Heike Heuer [6], Jens Mittag [4], Ruben Nogueiras [2], Markus Schwaninger [1] & Helge Müller-Fielitz [1] ✉

Several neuronal populations in the hypothalamus and brainstem express thyrotropin-releasing hormone (TRH). While TRH neurons in the paraventricular nucleus (PVN) regulate the thyroid axis, the roles of other TRH-producing neurons remain largely unknown. Here we investigate the role of TRH neurons in the PVN, the dorsomedial hypothalamus (DMH), the medial preoptic area (MPA), and the rostral raphe pallidus (RPa) for metabolism in mice. Selective activation of these populations using chemogenetics in mice revealed that TRH neurons of the hypothalamus increase food intake and influence energy homeostasis in different ways. Specifically, TRH neurons in the PVN and DMH enhance brown adipose tissue activity via a polysynaptic circuit, while MPA-located neurons increase locomotor activity and maintain cold tolerance. These effects were independent of the thyroid axis, demonstrating that TRH neurons have distinct, subtype-specific ways to increase energy expenditure beyond regulating the thyroid axis in mice.

Energy homeostasis entails the precise control of both energy intake and energy expenditure (EE). This balance is crucial for maintaining a constant energy supply in mammals, including humans, and is essential for preserving a stable body temperature. In addition to basal thermogenesis, adaptive mechanisms exist to maintain a constant body temperature. The brown adipose tissue (BAT) is one of the main thermogenic organs, primarily responsible for non-shivering thermogenesis (for review see refs. 1,2). BAT activity has been initially described as an essential defense mechanism against cold stress, infants and small organisms, with its importance declining in humans during adulthood[3,4].

Since the early 2000s, however, it has been established that the adult humans retain active BAT in the subclavicular region[5,6]. Elucidating the mechanisms involved in regulating EE and BAT-driven non-shivering thermogenesis may offer new opportunities to counteract metabolic disorders observed in obesity and diabetes[7,8].

The neuropeptide thyrotropin-releasing hormone (TRH) has been studied not only for its functions in the hypothalamic-pituitary-thyroid (HPT) axis, but also for its centrally mediated stimulatory effects on thermogenesis (for review[9]) and feeding behavior (for review[10]). Most studies on TRH-driven influence on energy metabolism have focused on

[1]Institute of Experimental and Clinical Pharmacology, Center of Brain, Behavior and Metabolism (CBBM), University of Lübeck, Lübeck, Germany. [2]Department of Physiology, CIMUS, University of Santiago de Compostela-Instituto de Investigación Sanitaria, Santiago de, Compostela, Spain. [3]Department Virus Immunology, Leibniz Institute of Experimental Virology (LIV), Hamburg, Germany. [4]Institute of Experimental Endocrinology, Center of Brain, Behavior and Metabolism, University of Lübeck, Lübeck, Germany. [5]Univ. Lille, Inserm, CHU Lille, Laboratory of Development and Plasticity of the Neuroendocrine Brain, Lille Neuroscience & Cognition, UMR-S 1172, DISTALZ, EGID, Lille, France. [6]Department of Endocrinology, Diabetes and Metabolism, University Hospital Essen, University Duisburg-Essen, Essen, Germany. [7]These authors contributed equally: Andreea Constantinescu, Akila Chandrasekar. ✉e-mail: Helge.muellerfielitz@uni-luebeck.de

either systemic[11] or i.c.v.[12,13] delivery of TRH in rodent models. The metabolic TRH effects were attributed both to interactions with the sympathetic nervous system (SNS)[11,14] and the stimulation of thyroid hormones[15]. The HPT axis is controlled by the hypophysiotropic TRH neurons[16], residing in the paraventricular hypothalamus (PVN) and projecting to the median eminence (ME), where they release TRH into the hypophyseal portal system. TRH activates thyrotropic cells in the anterior pituitary gland by stimulating the TRH receptor (TRHR) and resulting in the release of thyroid-stimulating hormone (TSH). TSH then stimulates thyroid hormone (thyroxine, T4, and 3,3',5-tri-iodothyronine, T3) production in the thyroid gland. Thyroid hormones[17-19] and TSH[20] play a role in maintaining metabolic homeostasis, making the HPT axis an important top-down regulatory pathway. TRH as well as its receptors are also found in other (extra-) hypothalamic areas suggesting additional CNS functions of this neuropeptidergic signaling system[21,22]. In contrast to humans, rodents express two different forms of TRH receptors, the TRHR1 and TRHR2[21], whereby TRHR1 is the receptor responsible for TSH release[23] and mediation of central changes in behavior[24,25].

Remarkably, many TRH-positive neurons are present in key nuclei involved in metabolic regulation, including the medial preoptic area (MPA), the dorsomedial hypothalamic nucleus (DMH), and the raphe nuclei, particularly the raphe pallidus (RPa), in addition to the PVN. As the primary brain region for temperature regulation, the MPA integrates sensory inputs of both temperature elevation and cold exposure[26-28] and exerts control over the DMH through inhibitory projections[27]. The DMH is responsible for the activation of BAT through projections to the RPa and the SNS[29,30], which in turn control BAT thermogenesis[31]. However, it is unclear whether TRH neurons contribute to thermogenesis and energy metabolism through pathways independent of the HPT axis.

In this study, we demonstrate that the hypothalamic TRH neurons in the MPA, DMH, and PVN, exert a significant influence on food intake and energy metabolism. Conversely, TRH neurons of the rostral RPa do not appear to be involved in these processes. Furthermore, our results suggest that DMH- and PVN-located TRH neurons are (poly)synaptically connected to the BAT, thereby enhancing BAT thermogenesis. In contrast, activation of MPA-located TRH neurons did not alter BAT activity but these neurons were essential for defending body temperature during cold exposure. In particular, we demonstrate that the metabolic effects of TRH neurons in the PVN are mediated independently of TRHR1 and thus of the HPT axis.

## Results

### TRH neurons are connected to the interscapular BAT

The central application of TRH increases BAT activity[12,13]. To identify TRH neurons connected to the BAT, we employed a retrograde tracing approach, using a pseudorabies virus. After injecting a GFP expressing pseudorabies virus (PRV) into the interscapular BAT (Fig. 1a), we identified GFP-positive TRH neurons using RNAScope in situ hybridization against *Trh* mRNA. In some experiments, we labeled TRH neurons by injecting a Cre-dependent mCherry expressing adeno-associated virus (AAV) in the hypothalamus together with the PRV in the BAT of TRH-IRES-Cre mice (Fig. 1a). Confirming previous studies, we found PRV-transduced neurons in the spinal cord (Fig. 1b), the gigantocellular reticular nucleus (GiA), the nucleus of the solitary tract (NTS), the rostral ventrolateral reticular nucleus (RVL), the parapyramidal nucleus (PPy), the locus coeruleus (LC), and the periaqueductal gray (PAG, Fig. 1c–f and Supplementary Fig. 1a–g) of the hind- and midbrain[32,33]. In accordance with a former study in rats[34], we found that PRV-positive neurons in the RPa expressed *Trh* and (Fig. 1c–f) tryptophan hydroxylase (TPH, Supplementary Fig. 1e), a marker for serotonergic neurons. In situ hybridization against *Trhr1* mRNA showed that the PRV-positive neurons of the LC, a key nucleus regulating the autonomic nervous system, expressed *Trhr1* mRNA (Supplementary Fig. 1g). This indicates a close interaction between the TRH network and the autonomic nervous system[14,35]. Only a few *Trhr1*-positive PRV-traced neurons were found in the RPa and GiA

(Supplementary Fig. 1j, k), while *Trhr1* mRNA exhibited high expression levels within the facial nucleus (7 M, Supplementary Fig. 1h–j) suggesting a previously postulated interaction between TRH and motoneurons[36,37].

In the hypothalamus, we identified PRV-positive cells in the DMH (Fig. 1g), PVN (Fig. 1l), and MPA (Fig. 1q). All three brain regions contain TRH-positive neurons (DMH: Fig. 1h, PVN: Fig. 1m, MPA: Fig. 1r). In the DMH (Fig. 1i–k) and PVN (Fig. 1n–p), we identified PRV-positive TRH neurons, confirming a (poly)synaptic link between the TRH neurons of these two areas and the BAT. However, in the MPA, we could not retrogradely trace any TRH neurons (Fig. 1s–u).

### Activation of TRH neurons in the PVN increased energy metabolism

The PVN is known as a control hub for multiple functions such as the release of hypophysiotropic hormones, regulation of metabolic homeostasis, and feeding behavior. Our PRV retrograde tracing experiments identified the PVN as a major nucleus in the CNS-to-BAT network (Fig. 1l–p), with TRH neurons being part of this circuitry (Fig. 1n–p). To determine the role of the PVN-located TRH neurons (PVN^TRH) in energy homeostasis, we generated PVN^TRH-hM3D mice by injecting the Cre-dependent vector AAV-flex-hM3D-mCherry, carrying an activator DREADD (Designer Receptors Exclusively Activated by Designer Drugs[38]), into the PVN of TRH-IRES-Cre mice (Fig. 2a). mCherry-labeled neurons expressed cFos after clozapine-N-oxide (CNO) treatment, confirming the successful activation of the transduced TRH neurons in PVN^TRH-hM3D mice (Fig. 2b, Supplementary Fig. 2a–c, Supplementary Fig. 11). In line with the presence of hypophysiotropic and non-hypophysiotropic TRH neurons in the PVN[22], we found mCherry-positive projections of TRH neurons to the ME, the arcuate nucleus (ARC) and the DMH (Fig. 2c). Additionally, we showed that mCherry-positive projections to *Trhr1*- and *Trhr2*-positive brain regions, namely the lateral septum, the thalamus, and the mammillary nucleus, colocalized with cFos indicating neuronal activation in these regions (Supplementary Fig. 10a–c). CNO treatment of PVN^TRH-hM3D mice increased TSH (Fig. 2d), T4 (Fig. 2e) and T3 (Fig. 2f) plasma concentrations, confirming a robust activation of the hypophysiotropic TRH neurons.

Given the finding of a neuronal pathway linking PVN^TRH neurons with the BAT (Fig. 1l–p), we investigated whether PVN^TRH neurons can activate BAT thermogenesis via infrared (IR) thermography in PVN^TRH-hM3D mice. CNO, but not vehicle, increased BAT temperature (Fig. 2g, h). To test whether PVN^TRH neurons activate the BAT through the SNS, we administered the β3-adrenoceptor blocker SR59230A in parallel with CNO. SR59230A prevented the BAT activation induced by PVN^TRH neurons (Fig. 2h), indicating that PVN^TRH neurons activate the BAT via the SNS. Activation of β3-adrenergic receptors stimulates in BAT the phosphorylation and activity of hormone-sensitive lipase (HSL)[39]. Phosphorylation of HSL (pHSL) was upregulated 90 minutes (min) following the activation of PVN^TRH neurons and was normalized by SR59230A. This confirmed that the PVN^TRH neurons can stimulate BAT activity, probably by activating β3-adrenergic receptors (Fig. 2i).

To further characterize the physiological functions of PVN^TRH neurons, we conducted calorimetric measurements in PVN^TRH-hM3D mice (Fig. 2a). Activation of the PVN^TRH neurons rapidly increased energy expenditure (EE, Fig. 2j, k). The EE was significantly increased even after accounting for body weight as a covariate (Fig. 2l, Supplementary Fig. 2d, e). In parallel, the core body temperature was elevated compared to the i.p. injection of vehicle (Fig. 2m, n) or CNO in control mice (Supplementary Fig. 4c, d). While locomotion was not significantly elevated after activation of PVN^TRH neurons (Fig. 2o), CNO treatment led to increased food intake (Fig. 2p) during the light phase (Fig. 2q), with a normal feeding behavior during the following dark phase (Fig. 2r). The increase in *Fos* mRNA in the ARC confirms prior studies showing that TRH neurons of the PVN activate the orexigenic agouti-related peptide (AgRP) neurons[40,41]. However, PVN^TRH neurons do not regulate the expression of *Agrp, Npy* and *Pomc* mRNA (Supplementary Fig. 9). The increased food

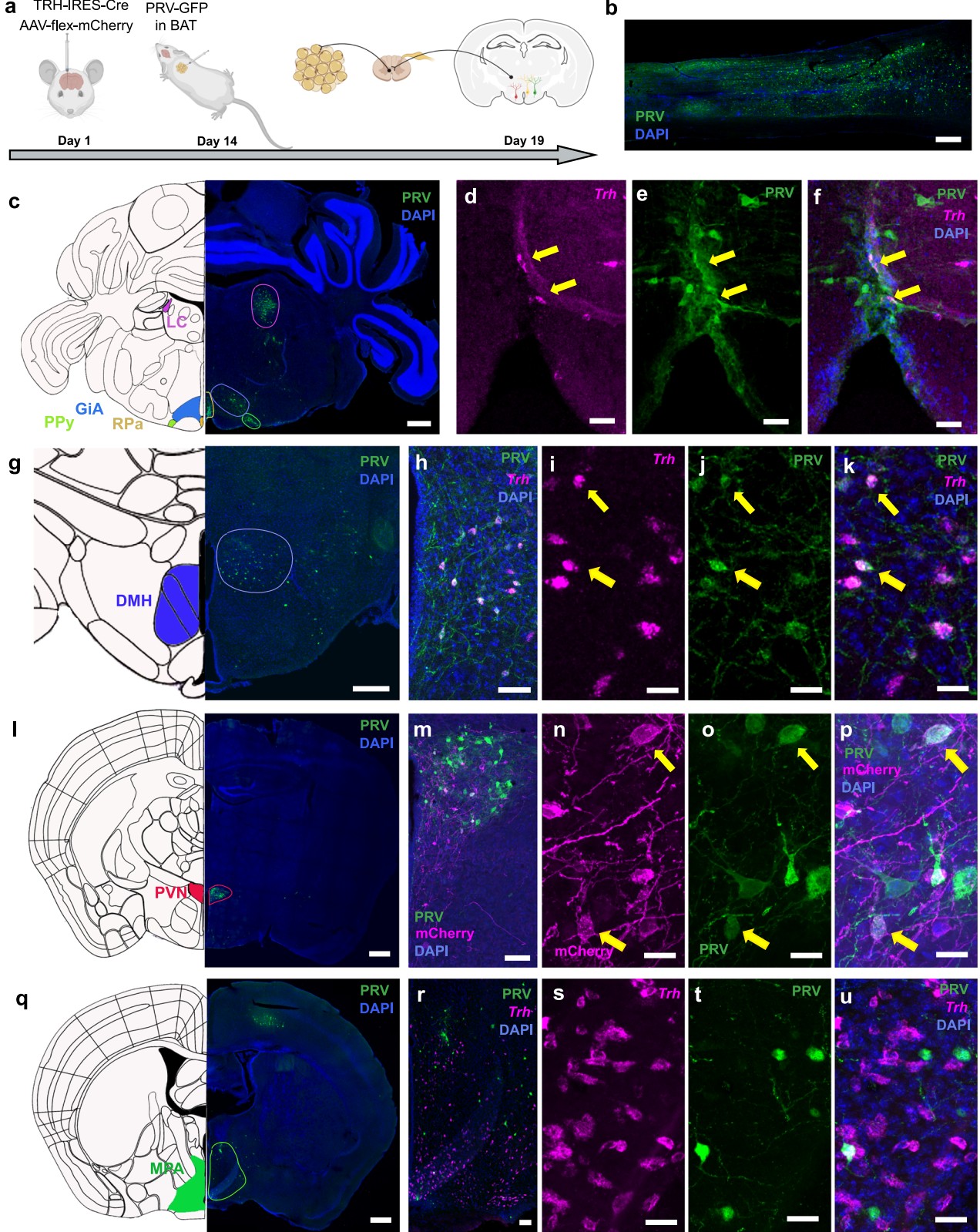

intake was associated with a higher RER, spanning over the entire light phase post-treatment (Fig. 2s), reflecting the metabolism of carbohydrates from the increased food consumption, as described after direct activation of AgRP neurons[42]. To test whether the increase in EE and core body temperature was mediated by increased food intake, we stimulated the PVN$^{TRH}$ neurons while fasting the animals during the 6 h after CNO injection (Supplementary Fig. 2f). Increase in EE and core body

temperature persisted, indicating a food-independent elevation (Supplementary Fig. 2g, h).

## TRH neurons in the DMH produce similar metabolic effects as those in the PVN

The DMH contains a substantial population of TRH neurons connected to the BAT (Fig. 1g–k). To examine the acute metabolic function of

**Fig. 1 | TRH neurons of the hypothalamus are interconnected with the BAT.**
**a** Schematic protocol of the experiment. TRH-IRES-Cre mice were injected with an AAV-flex-mCherry in the PVN thus labeling TRH positive neurons **b** Sagittal section of the spinal cord (T1-T7) 5 days after injection of a GFP expressing pseudorabies virus (PRV, green) into the iBAT. **c** Representative image of the hindbrain of PRV (green) injected mice including the PRV positive areas of the raphe pallidus (RPa, **d**–**f**), the gigantocellular reticular nucleus (GiA), the parapyramidal nucleus (PPy), and the locus coeruleus (LC). *Trh* mRNA (magenta, **d** and the PRV (green, **e** were colocalized (**f**, arrows) in the RPa. GFP-positive cells were found in the dorsomedial hypothalamus (DMH, **g**–**k**), paraventricular hypothalamus (PVN, **l**–**p**), and the

medial preoptic area (MPA, **q**–**u**). Colocalization of *Trh* mRNA (magenta, **i**) with PRV (green, **j**) in the DMH (**h**). In the PVN (**l**, **m**), TRH positive cells, labeled by mCherry, (**n**) colocalized (**p**, arrows) with PRV (green, **o**). In the MPA (**q**, **r**) *Trh* mRNA (RNAScope, magenta, **s**) did not colocalize with PRV (green, **t**; **u**). Scalebars: **b**, **c**, **g**, **l**, **q**: 500 μm; **h**, **m**, **r**: 100 μm; **d**, **e**, **f**, **i**, **j**, **k**, **n**, **o**, **p**, **s**, **t**, **u**: 20 μm. All staining and RNAScopes were repeated at least 2 times in 3 independent animals. Parts of **a** created in BioRender. Schwaninger, M. (2026) https://BioRender.com/gtmrs40. Drawings in e, g, l, and q were adapted from the Allen Reference Atlas – Mouse Brain [Adult Mouse]. Available from atlas.brain-map.org[71].

DMH-located TRH neurons (DMH[TRH]), we administered AAV-flex-hM3D-mCherry into the DMH of TRH-IRES-Cre mice, thereby generating DMH[TRH]-hM3D mice (Fig. 3a, Supplementary Fig. 11). CNO treatment stimulated cFos expression in DMH[TRH] neurons confirming correct targeting of the nucleus and successful activation of the neurons (Fig. 3b, c, and Supplementary Fig. 3a–c). Notably, we did not observe any mCherry-positive TRH neuronal projections in the ME, indicating that only non-hypophysiotropic TRH neurons were transduced (Fig. 3c). In line with the absence of transduced hypophysiotropic TRH neurons, CNO treatment did not change plasma levels of TSH (Fig. 3d), T4 (Fig. 3e), and T3 (Fig. 3f). During calorimetry, activation of DMH[TRH] neurons significantly increased EE (Fig. 3g–i and Supplementary Fig. 3d, e), raised core body temperature (Fig. 3j, k), and elevated BAT activity, as indicated by a significant increase in BAT temperature (Fig. 3l, m). Similar to PVN[TRH] neurons, DMH[TRH] neurons activated BAT through the SNS, since in DMH[TRH]-hM3D mice the CNO-induced increase in BAT temperature was reduced (Fig. 3m) and pHSL was blocked by SR59230A (Fig. 3n). CNO treatment did not change home cage locomotion (Fig. 3o). Similar to PVN[TRH]-hM3D mice, we observed a significant increase in food consumption in DMH[TRH]-hM3D mice after CNO treatment (Fig. 3p–r). This effect occurred transiently after the acute treatment during the light phase (Fig. 3q). As expected, the increased food intake was accompanied by a significant increase in *Fos* in the ARC (Supplementary Fig. 9) and RER (Fig. 3s). The modulation of EE and body temperature following stimulation of the DMH[TRH] neurons was also independent of increased food intake (Supplementary Fig. 3f–h). Thus, activation of DMH[TRH] neurons exhibited the same metabolic profile as PVN-located neurons, but without any impact on the HPT axis.

### TRH neurons of the RPa did not affect energy metabolism
As a third TRH population connected to BAT, we investigated the TRH neurons in the rostral RPa (RPa[TRH]). The RPa plays a role in temperature regulation during cold exposure[2]. We identified RPa-located TRH neurons to be (poly)synaptically connected to the BAT using the PRV approach (Fig. 1c–f). Injections into the rostral RPa and the proper transduction with the AAV were validated by mCherry staining (Fig. 4a). CNO treatment of RPa[TRH]-hM3D mice effectively activated the transduced TRH neurons, as shown by colocalization of mCherry and cFos (Fig. 4b). Compared to vehicle treatment, activation of RPa[TRH] neurons with CNO resulted in no changes in EE (Fig. 4c, d) or home cage locomotion (Fig. 4e). Neither BAT temperature, measured by IR thermography, nor food intake or RER were affected by the CNO treatment (Fig. 4f–j). These data were comparable with mCherry transduced controls (Supplementary Fig. 4). This indicates that the transduced TRH neurons in the rostral RPa do not influence the regulation of energy metabolism or food intake.

### TRH neurons of the MPA increase locomotion but not BAT activity
Given the role of the MPA in temperature regulation, we also investigated the function of the TRH neurons in this area (MPA[TRH]). However, in contrast to the TRH neurons in the PVN, DMH, and RPa, these neurons were not connected to the BAT (Fig. 1q-u). To generate the MPA[TRH]-

hM3D mice, we injected the Cre-dependent AAV-flex-hM3D-mCherry vector into the MPA of TRH-IRES-Cre mice (Fig. 5a-c, Supplementary Fig. 5a-c, and Supplementary Fig. 11). We observed mCherry-positive projections to the DMH and ARC, but none into the ME (Fig. 5c). Accordingly, activation of the transduced MPA[TRH] neurons did not activate the HPT axis, as indicated by the unchanged levels of TSH (Fig. 5d), T4 (Fig. 5e), and T3 (Fig. 5f) following CNO treatment. Nevertheless, the activation of MPA[TRH] neurons led to a significant, food- and bodyweight-independent increase in EE compared to vehicle treatment (Fig. 5g–i and Supplementary Fig. 5d–h). This was accompanied by an increase in core body temperature (Fig. 5j, k) which, however, was significantly reduced after 5 hours of food restriction compared to the CNO stimulation with *ad libitum* access to food (Supplementary Fig. 5g). However, the rise in core body temperature was not associated with a thermogenic BAT response (Fig. 5l, m). Western blot analysis of the pHSL/HSL ratio in the BAT confirmed this observation (Fig. 5n). In contrast to the other investigated TRH neuron populations, the activation of MPA[TRH] neurons significantly increased the home cage locomotion of mice (Fig. 5o, p). Interestingly, the stimulation of the MPA-located TRH neurons also significantly elevated food intake (Fig. 5q), which appears to be a common function of the hypothalamic TRH neurons in the MPA, DMH, and PVN (Supplementary Fig. 9). The orexigenic response was short-lasting, only occurred acutely during the light phase (Fig. 5r), and was accompanied by an increase in RER (Fig. 5t), with nighttime feeding being unaffected (Fig. 5s).

### Metabolic effects of taltirelin are TRHR1-dependent
TRH functions as a neurotransmitter by activating the two specific TRH receptors TRHR1 and TRHR2 in mice. To investigate their contribution to the metabolic effects observed, we administered the TRH-analog taltirelin[43] or vehicle via intraperitoneal injection in *WT*, *Trhr1*[-/-], and *Trhr2*[-/-] mice using a crossover design on separate days. Taltirelin treatment increased EE in *WT* and *Trhr2*[-/-] mice, an effect that was absent in *Trhr1*[-/-] mice (Fig. 6a–d and Supplementary Fig. 6a–g). The increase in EE was accompanied by a TRHR1-dependent increase in locomotor activity (Fig. 6e–h), body temperature, and BAT activation as investigated by BAT surface temperature and pHSL levels (Fig. 6i–l).

Via TRHR1 in the pituitary[21], taltirelin significantly increased plasma levels of TSH, T4, and T3 in *WT* mice. The TSH release stimulated by taltirelin treatment was abolished in TRHR1 knockout mice (Supplementary Fig. 6p–r). These experiments demonstrated a TRHR1-dependent regulation of EE, body temperature, BAT thermogenesis, HPT axis and locomotor activity by taltirelin.

Taltirelin caused a slight increase in food intake during the light phase in *WT* mice compared to *Trhr1* and *Trhr2* knockout mice also injected in the light phase, instead of reducing food intake as expected[10] (Supplementary Fig. 6h–k). However, over the course of the day, all taltirelin-injected mice showed reduced food intake during the following dark phase compared to the respective vehicle-treated animals (Supplementary Fig. 6l).

Despite the increased food ingestion, the RER was reduced after taltirelin treatment in *WT* mice (Supplementary Fig. 6m–o), indicating the use of lipids as an energy source. Taltirelin may exert a peripheral effect on the availability of lipids and food intake.

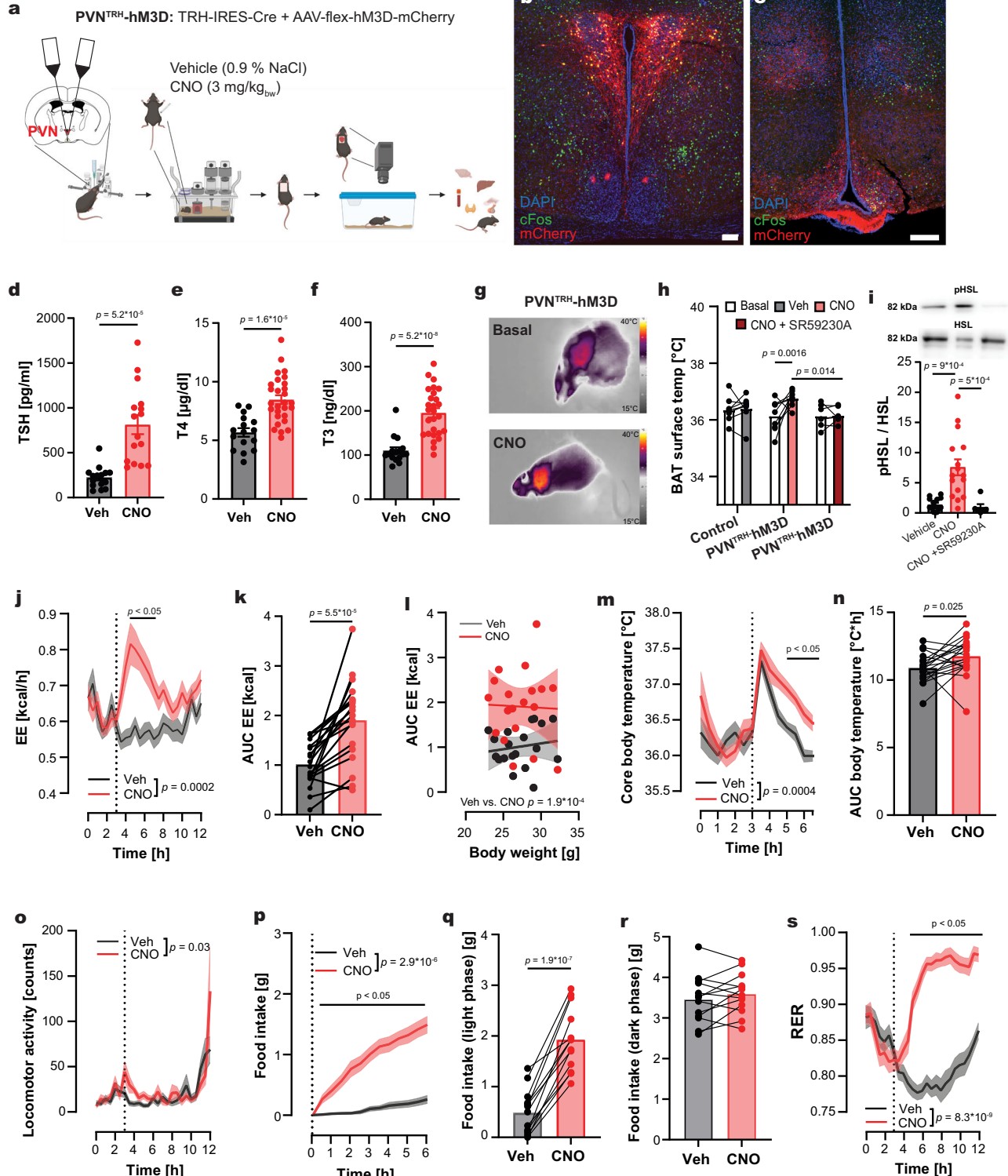

## Metabolic effects of PVN-located TRH neurons were TRHR1-independent

After elucidating the TRHR1-dependent taltirelin effects on the HPT axis and metabolic parameters, we proceeded to investigate the role of the HPT axis and the TRH receptors in the PVN$^{TRH}$-hM3D model. To this end, we crossed TRH-IRES-Cre mice with *Trhr1*$^{-/-}$ and *Trhr2*$^{-/-}$ mice to generate two single knockout lines (*Trhr1*$^{-/-}$:TRH-IRES-Cre, *Trhr2*$^{-/-}$:TRH-IRES-Cre) and a double knockout line (*Trhr1*$^{-/-}$:*Trhr2*$^{-/-}$:TRH-IRES-Cre) carrying the Cre-recombinase in TRH neurons. Animals were injected with AAV-flex-hM3D-mCherry into

the PVN and underwent calorimetry and thermography analyses as previously described.

Activation of TRH neurons in the PVN increased TSH, T4, and T3 plasma concentrations in *Trhr2*$^{-/-}$:PVN$^{TRH}$-hM3D mice to the same extent as in *WT*:PVN$^{TRH}$-hM3D mice. In contrast, due to the lack of functional TRHR1, *Trhr1*$^{-/-}$:PVN$^{TRH}$-hM3D, and *Trhr1*$^{-/-}$:*Trhr2*$^{-/-}$:PVN$^{TRH}$-hM3D mice did not respond to the activation of the PVN-located TRH neurons with an increase in these hormone levels (Fig. 7a–c). In TRHR1 knockout mice, TSH, T4, and T3 plasma concentrations were similar, after stimulation, to baseline levels of littermate controls (Fig. 7a–c).

**Fig. 2 | Effects on energy homeostasis after activation of TRH neurons in the PVN. a** Stereotactic injection of AAV-flex-hM3D-mCherry into the PVN of TRH-IRES-Cre mice generated PVN[TRH]-hM3D mice. **b, c** Representative immunostaining for mCherry (red) and cFos (green) in the PVN (**b**) and ME (**c**). Nuclei were stained with DAPI (blue). Scale bar: 100 μm. Staining was repeated in 10 animals. **d–f** Plasma levels of TSH (**d**), T4 (**e**), and T3 (**f**) 90 min after treatment with CNO (red) or vehicle (gray, Veh). **g** Representative IR-thermography images before and CNO. **h** Changes in BAT surface temperature before (white) and after treatment with vehicle (gray), CNO (red), or CNO + SR59230A (dark red). **i** Representative western blots for pHSL and total HSL in BAT 90 min after treatment with vehicle, CNO, or CNO + SR59230A (membranes shown in Supplementary Fig. 12). Quantification of HSL phosphorylation as ratio of pHSL/HSL. **j, k** Changes in energy expenditure (EE, **j**) and as area under the curve (AUC, **k**) after CNO (red) or vehicle (gray) treatment. **l** Linear regression of changes in AUC$_{EE}$ from j against the body weight after vehicle (gray)

and CNO (red) treatment. **m, n** Changes in core body temperature (**m**) and corresponding AUCs (**n**) following CNO or vehicle injection. **o** Changes in home cage locomotor activity after CNO (red) or vehicle (gray) treatment. **p–r** Food intake following CNO versus vehicle injection (**p**), with quantification of total intake during the light (**q**) and dark (**r**) phases. **s** Respiratory exchange ratio (RER) curves. All mice treated with CNO or vehicle during the light phase in a crossover design. Data are presented as mean ± s.e.m. For (**d–f, k, n, q, r**), two-tailed unpaired t-tests; for (**h, j, o, p, s**), two-way RM ANOVA followed by Šídák's multiple-comparisons test; for m, a REML mixed-effects model followed by Šídák's multiple-comparisons test; for i, a two-sided Brown–Forsythe and Welch ANOVA followed by Dunnett's T3 multiple-comparisons test; for l, one-way analysis of covariance (ANCOVA) was used. Exact p-values, sample sizes and statistical details are provided in Supplementary Table 1. Source data are provided with this paper. Parts of figure a created in BioRender. *Schwaninger, M. (2026)* https://BioRender.com/xbrkg90.

Despite the abrogated HPT activation, stimulation of TRH neurons in the PVN of *Trhr1*[-/-] animals significantly increased EE (Fig. 7d), like *WT*:PVN[TRH]-hM3D mice (Fig. 7g), suggesting that the TRHR1 does not play a significant role in elevating EE. The absence of a functional TRHR2 inhibited the effect with a flattening of the EE curve (Fig. 7e–g, and Supplementary Fig. 7a–g) indicating an involvement of the TRHR2 in the EE increase induced by the TRH neurons located in the PVN. The TRH neurons located in the PVN projected, among other regions, to nuclei that expressed *Trhr2* mRNA and further showed cFos activation upon CNO stimulation (Supplementary Fig. 10) suggesting their involvement in flattening the EE. Surprisingly, we observed an increase in body temperature and BAT activity after CNO stimulation in animals lacking TRHR1 and TRHR2. The activation of TRH neurons in the PVN increased core body temperature independent of the TRH receptors (Fig. 7h–k). Likewise, activation of TRH neurons in the PVN significantly increased the BAT temperature independent of TRHR1 and TRHR2 (Fig. 7l–o).

As TRH neurons of the PVN induced feeding (Fig. 2p, q) and previous studies have demonstrated that this effect is facilitated by the co-release of glutamate from the TRH population acting on AgRP neurons[40,41], we investigated whether the orexigenic effect of TRH neurons in the PVN depends on TRHR1 and TRHR2. The stimulation of the TRH neurons in the PVN of TRHR1, TRHR2, and double knockout mice by CNO led to a robust increase in food intake compared to the vehicle treatment (Fig. 7p–r) and similar to *WT* mice (Fig. 7s). Taken together, these findings suggest that the increase of the BAT thermogenesis and feeding behavior, while controlled through the TRH neurons of the PVN, was not dependent on TRHR1 or TRHR2 and took place independently of the hormones of the HPT axis.

### Influence of chronic inactivation of hypothalamic TRH neurons in acute cold response

After lowering the ambient temperature from 23 °C to 10 °C for 4 h, we detected *Fos* positive TRH neurons in the PVN, DMH, and MPA (Supplementary Fig. 8a–i), indicating that TRH neurons in these regions are responsive to acute cold exposure. The activation of hypophysiotropic TRH neurons was additionally identified due to increased T3 levels (Supplementary Fig. 8o, p). To investigate the relationship between cold-activated TRH neurons and the increase in EE, we performed an acute cold tolerance test while chronically blocking TRH neuron transmission with tetanus toxin. Following transduction of PVN[TRH], DMH[TRH], and MPA[TRH] neurons with AAV-Syn1-flex-TeNT-2A-[nuc]TdTomato[44] (Fig. 8b), PVN[TRH]-TeNT mice showed an increase in body weight compared to AAV-CAG-flex-mCherry transduced control mice over 14 days (Fig. 8c). In contrast, five days after transduction, MPA[TRH]-TeNT mice rapidly lost body weight, which stabilized after two days (Fig. 8c). This change in MPA[TRH]-TeNT mice was associated with an increased lean/fat mass ratio (Fig. 8d) and a trend towards lower fat mass (Supplementary Fig. 8j–l). Additionally, these body weight

changes were not associated with changes in food intake in either the PVN[TRH]-TeNT or MPA[TRH]-TeNT groups. However, similar to the findings of Douglass et. al[45]., DMH[TRH]-TeNT mice showed an alteration in circadian food intake, characterized by a reduction in food consumption during the dark phase and a corresponding increase during the light phase, such that 24-hour food intake was comparable in all groups (Supplementary Fig. 8m, n).

A rapid reduction of ambient temperature from 23 °C to 10 °C resulted in a pronounced drop in body temperature in MPA[TRH]-TeNT mice compared to control mice (Fig. 8e, f). During cold exposure, a milder decrease in body temperature was observed in DMH[TRH]-TeNT mice, whereas PVN[TRH]-TeNT mice showed no reduction at all (Fig. 8e, f). The strong decline in body temperature observed in MPA[TRH]-TeNT mice was associated with an inability to increase EE, a response that was evident in all other groups upon cold exposure (Fig. 8g–j). This finding was confirmed even when lean body mass was included as a covariate (Fig. 8k–n) and was not due to a change in thyroid hormone levels (Supplementary Fig. 8o, p).

## Discussion

The present study provides evidence that different populations of TRH neurons in the hypothalamus regulate energy homeostasis. Interestingly, the TRH populations differed in the parameters and the physiological functions they influenced. Remarkably, activation of all three hypothalamic TRH neuron populations stimulated food intake, increased energy expenditure, and body temperature, revealing common characteristics of these TRH neurons. Activation of TRH neurons located in the PVN and DMH resulted in increased thermogenic activity of the BAT, a finding supported by the (poly)synaptic connection between these populations and the BAT. In contrast, this effect could not be attributed to TRH neurons located in the MPA, which are also not connected to the BAT. The activation of MPA-located TRH neurons appears to increase locomotor activity with an associated increase in energy consumption. Further, these neurons are essential for maintaining body temperature during acute cold exposure. Only activation of the TRH neuron population in the PVN increased plasma levels of TSH, T4, and T3, in a TRHR1-dependent manner. However, the effects on energy metabolism mediated by PVN-located TRH neurons were TRHR1 independent thus indicating a mechanism separate from thyroid hormone release. The examined TRH neurons located in the rostral RPa had no role in controlling food intake or energy homeostasis. Our data show the physiological functions and differences of TRH-expressing neuronal populations in regulating energy metabolism.

The centrally mediated activation of BAT thermogenesis is an important component of thermoregulation. Cold exposure leads to rapid activation of TRH neurons in the PVN and RPa[46]. Here we show that in addition to PVN[TRH] neurons, TRH neurons in the MPA and DMH are also activated after 4 hours of cold exposure. Furthermore,

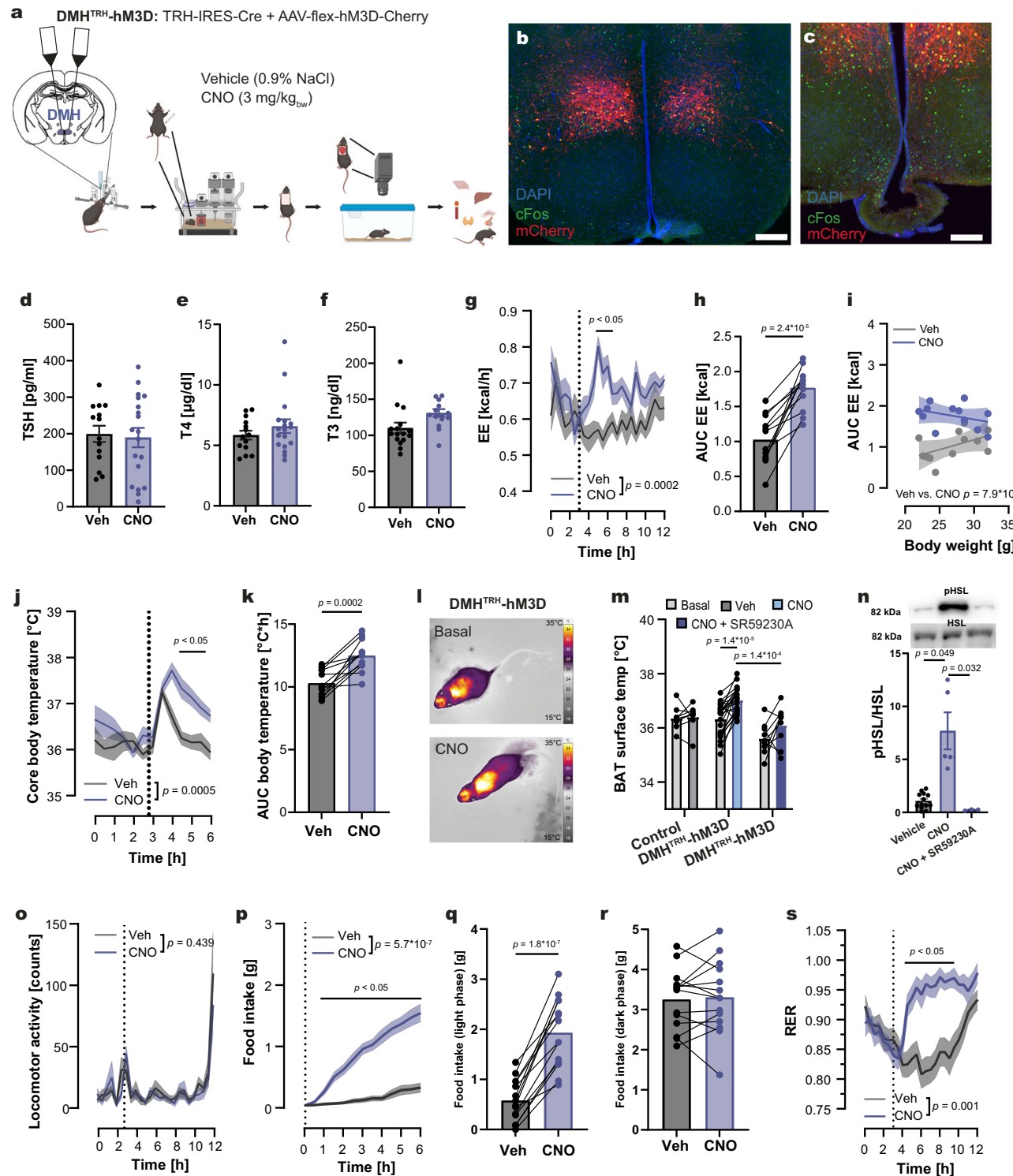

retrograde tracing provides evidence that TRH neurons in the PVN and DMH are an integral part of the central network regulating BAT activity. Interestingly, the TeNT expression in PVN$^{TRH}$ neurons does not impair the acute cold response, whereas inactivation of TRH neurons in the DMH and MPA markedly compromises this response. The cold-induced increase in plasma T3 levels was not altered by inactivating the PVN$^{TRH}$ neuron population, suggesting ineffective inhibition of hypophysiotropic TRH release by tetanus toxin. It is also possible that reduced negative feedback at the level of tanycytes may mask potential effects on the HPT axis[47,48]. The MPA receives cold-related sensory inputs and indirectly stimulates the RPa, which projects to autonomic

nuclei in the spinal cord[2]. However, based on our findings, we conclude that neither the thermosensitive neurons in the MPA nor the neurons activated along the MPA-DMH-RPa-BAT pathway are likely to be TRH-positive. Nevertheless, inactivation of MPA$^{TRH}$ neurons significantly affects body weight and causes cold intolerance, reflecting their inability to generate sufficient energy to maintain body temperature. Thus, MPA$^{TRH}$ neurons represent an important component in the regulation of energy homeostasis.

Interestingly, many PRV-positive neurons connected to the BAT express *Trhr1* (as seen via RNAScope). The responsiveness of neurons in the LC and NTS to TRH has been reported before, albeit with no

**Fig. 3 | Effects on energy homeostasis after activation of TRH neurons in the DMH. a** Stereotactic injection of AAV-flex-hM3D-mCherry into the DMH of TRH-IRES-Cre mice generated DMH[TRH]-hM3D mice. **b, c** Immunohistochemical validation by mCherry (red) and DREADD activation by cFos staining (green) 90 min after CNO in the DMH (**b**) and ME (**c**) of DMH[TRH]-hM3D mice. Nuclei were counterstained with DAPI (blue). Scale bar: 100 μm. Staining was repeated in 10 animals. **d, f** Plasma levels of TSH (**d**), T4 (**e**), and T3 (**f**) measured 90 min after CNO (blue) or vehicle (gray, Veh) treatment. **g, h** Energy expenditure (EE, **g**) and corresponding AUCs (**h**) following CNO (blue) or vehicle (gray) treatment. **i** Linear regression of the individual changes of the AUC$_{EE}$ from g against the body weight after vehicle (gray) and CNO (blue) treatment. **j, k** Core body temperature (**j**) and AUC analysis (**k**) after vehicle or CNO treatment. **l** Representative IR-thermography images before (basal) and 45 min after CNO injection. **m** Individual changes in BAT surface temperature before (light gray) and after vehicle (dark gray), CNO (blue), or CNO + SR59230A (dark blue) treatment. **n** Representative Western blots of pHSL and total HSL in BAT 90 min after vehicle, CNO, or CNO + SR59230A treatment (membranes in Supplementary Fig. 12). Quantification of HSL phosphorylation is shown as ratio of pHSL/HSL for each group. **o** Home cage locomotor activity after treatment with CNO (blue) or vehicle (gray). **p–r** Food intake following CNO or vehicle injection over time (**p**), and individual total intake during the light (**q**) and dark (**r**) phases. **s** Respiratory exchange ratio (RER) curves during the light phase in a crossover design. Data are presented as mean ± s.e.m. For (**d–f, h, k, b, r**), two-tailed unpaired t-tests; for (**g, j, m, o, p, s**), two-way RM ANOVA followed by Šídák's multiple-comparisons test; for n, a two-sided Brown–Forsythe and Welch ANOVA followed by Dunnett's T3 multiple-comparisons test; for i, one-way analysis of covariance (ANCOVA) was used. Exact p-values, sample sizes and statistical details are provided in Supplementary Table 1. Parts of figure a created in BioRender. *Schwaninger, M. (2026)* https://BioRender.com/xbrkg90.

reference to the source of TRH[14,35]. Notably, TRH receptors are widely spread in the brain and spinal cord[21,49] and multiple brain regions have been demonstrated to be responsive to TRH[50–53] which indicates that several pathways could be involved in BAT activation. TRH and its analogs have been administered both centrally and peripherally[12,54], in both cases treatment-induced thermogenesis led to TRH being considered an essential element in counteracting hypothermia[55]. Additionally, intravenous injection of TRH increases BAT activity in humans under cold conditions, as shown by fMRI[56]. We show that i.p. injection of taltirelin led to a TRHR1-dependent increase in BAT temperature, core body temperature, locomotor activity, and consequently in EE of mice[12].

The stimulation of TRH neurons in the PVN replicated the effects observed with acute taltirelin treatment, except for locomotor activity and TRHR1 dependence. The PVN contains several TRH-positive neuronal populations[57]. The fact that only PVN[TRH]-hM3D mice, expressing a functional TRHR1, responded with elevated plasma levels of TSH, T4, and T3 after CNO treatment confirms the postulated TRHR1 dependency of the HPT axis[21,23]. The HPT axis is an important mechanism for the cold response, with TRH neurons being activated by cold exposure[58,59], mediating the release of TSH and thyroid hormones, which induces BAT thermogenesis[18,60]. However, the thermogenic effect of thyroid hormones is more delayed[61] and does not align with the rapid stimulation of BAT activity and EE observed with the activation of PVN[TRH] neurons. Nevertheless, gene regulation and activity in BAT result from a multifactorial interaction between sympathetic nervous system activity and endocrine factors such as thyroid hormones. Especially thyroid hormones are considered important regulators of the BAT activity. A prolonged increase in thyroid hormone levels is known to directly raise body temperature and promote BAT mass and lipid deposition, without affecting BAT temperature itself[62,63]. Together with our observation that the metabolic effects of PVN[TRH] neurons persist in animals with an inactivated HPT axis due to a *Trhr1* knockout, this suggests that rapidly changing thyroid hormone levels is less relevant for the acute and short-term regulation of BAT. It appears that TRH neurons regulate thermogenesis primarily through central mechanisms such as the SNS, rather than through elevated thyroid hormone levels. This mechanism may represent an additional central pathway contributing to the acute thermogenic activation of BAT in newborns, independent of thyroid hormones[4]. In addition, our data suggest that TRHR2 contributes to the modulation of PVN[TRH] neuron-mediated increases in EE, without affecting BAT activity or core body temperature. We identified a functional connection between non-hypophysiotropic PVN[TRH] neurons and downstream regions such as the lateral septum, mammillary nucleus, and thalamus, where increased *Fos* expression in *Trhr2*-positive neurons was observed following PVN[TRH] neuron stimulation. Notably, these brain regions have recently been associated with the regulation of food intake[52,64] and may also be involved in the modulation of energy

metabolism. These findings provide insights into the function of non-hypophysiotropic TRH neurons in the PVN. Considering that thermoregulation was not affected by the absence of TRHR1 or TRHR2, we suggest that the main effect of non-hypophysiotropic TRH neurons in the PVN exert their role through glutamate signaling, a commonly co-released neurotransmitter of TRH neurons[40,41]. Although increased food intake can enhance thermogenesis and EE[65,66], the TRH neuron-mediated increase in EE and core body temperature occurs at least partially independent of food intake, as the effect persisted even when the animals were fasted during TRH neuron activation.

An increase in locomotion is a specific characteristic of MPA[TRH] neuron stimulation. At the same time, we observed an increase in core body temperature and EE, but no change in BAT activity. Activation of MPA[TRH] neurons mediates a similar increase in motor activity as taltirelin, probably through dopamine release[67]. We propose that both the elevated core body temperature and EE in MPA[TRH]-hM3D mice result from increased locomotion and possibly muscular activity. The chronic inactivation of MPA[TRH] neurons led to a drastic reduction in body weight and cold intolerance. MPA[TRH]-TeNT mice were unable to increase their energy consumption at lowered ambient temperatures, which led to a reduced body temperature. This indicates an important function of MPA[TRH] neurons in the processing of cold stimuli. To date, other studies inhibiting the entire MPA through pharmacological drug applications or lesions have shown increases in locomotion and energy metabolism[68–70], suggesting that MPA[TRH]-mediated excitatory pathways coexist with inhibitory mechanisms of locomotion in the MPA.

The orexigenic effect of all investigated hypothalamic TRH neuron populations indicates a common function of these neurons. Previous work identified orexigenic glutamatergic projections of PVN-located TRH neurons into the ARC, which activates AgRP neurons and leads to a fast increase in food intake[40]. The independence of the orexigenic effect from TRH receptors confirms previous investigations showing that TRH does not affect the membrane potential or spontaneous spiking of POMC and NPY neurons[53]. This reinforces the observation that the TRH neurons act on the AgRP neurons via glutamatergic synapses, and lead to long-term changes in food intake by modulating glutamatergic activity[41]. Additionally, TRH and its analogs are also known to induce a quick reduction of food intake in several species and treatment protocols[10]. While taltirelin does not change food intake in the first hours after treatment, we do observe an increase at the end of the light phase which is contrary to previous reports[10]. As the acute anorexic effect of TRH is short (up to one hour) and our studies were conducted in the light phase, when basal food intake is low, we may have missed this effect. The importance of TRH neurons in the regulation of feeding behavior was further highlighted by alterations in the circadian pattern of food intake following chronic inhibition of DMH[TRH] neurons via suppression of neuronal transmission. These mice (DMH[TRH]-TeNT) consumed equal amounts of food during the light and dark phases, without changes in total 24 h food

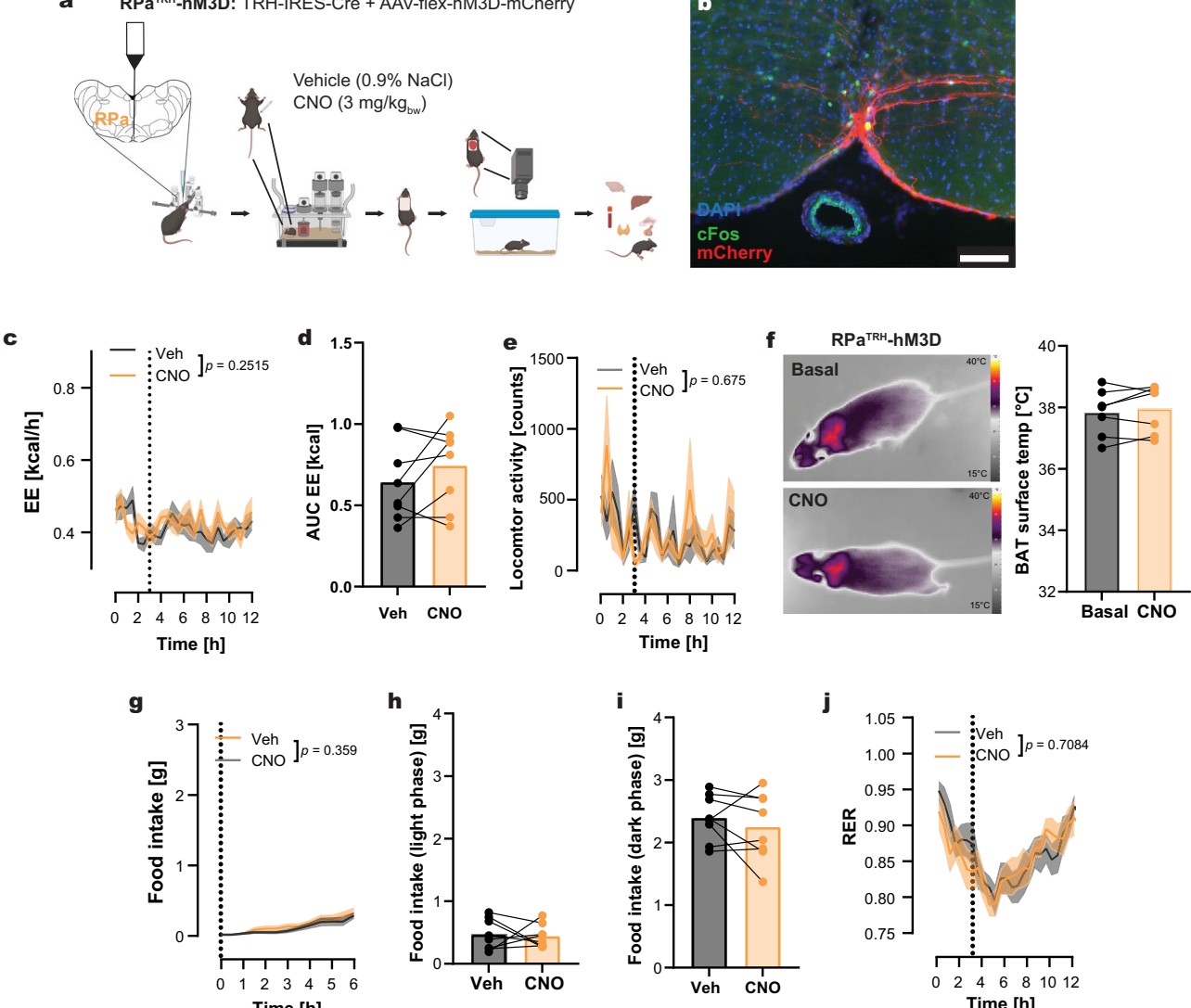

**Fig. 4 | Activation of TRH neurons in the RPa does not affect energy metabolism. a** Stereotactic injection of AAV-flex-hM3D-mCherry into the RPa of TRH-IRES-Cre mice generated RPa^TRH-hM3D mice. **b** Immunohistochemical validation of injection sites by mCherry expression (red) and confirmation of DREADD activation by cFos staining (green) 90 min after CNO (orange) injection. Nuclei were stained with DAPI (blue). Scale bar: 100 μm. Staining was repeated in 8 animals. **c**, **d** Energy expenditure (EE) over time after CNO (orange) and vehicle (gray) treatment (**c**), and corresponding AUCs (**d**). **e** Home cage locomotor activity over time following CNO (orange) or vehicle injection. **f** Individual changes in BAT surface temperature before (gray) and after CNO treatment (orange), assessed by IR-thermography.

**g** Cumulative food intake in a crossover design comparing CNO and vehicle treatment. **h, i** Individual changes in total food intake during the light (H) and dark (I) phases under vehicle or CNO conditions. **j** Respiratory exchange ratio (RER) curves during the light phase in mice treated with CNO or vehicle in a crossover design. All data are presented as mean ± s.e.m. For (**d, f, h, i**), two-tailed unpaired t-tests were used. For (**c**) and (**g**) two-way RM ANOVA followed by Šídák's multiple-comparisons test was used. For (**e**) and (**j**) a REML mixed-effects model followed by Šídák's multiple-comparisons test was used. Exact p-values, sample sizes and statistical details are provided in Supplementary Table 1. Parts of figure a created in BioRender. *Schwaninger, M. (2026)* https://BioRender.com/xbrkg90.

intake, confirming the findings of Douglass et al. using the same animal model[45].

In conclusion, all the investigated hypothalamic TRH neuron populations play a pivotal role in regulating energy metabolism, with each exerting the common effect of increased food intake. However, the regulatory patterns differ between TRH populations, leading to variations in metabolic outcomes. For example, MPA^TRH neurons specifically increase locomotor activity, modulate body weight and cold response, while DMH- and PVN-located neurons induce BAT activity. Furthermore, only the PVN contains hypophysiotropic TRH neurons that stimulate TSH release in a TRHR1-dependent manner. In contrast, the metabolic effects of PVN-located TRH neurons are independent of TRHR1 and thus of the hormones of the HPT axis, contrasting with the

TRHR1-dependent metabolic effects of taltirelin. This suggests the existence of additional TRH-dependent regulatory circuits that are independent of PVN-located TRH neurons.

## Methods

### Animal care

All animal experiments were conducted in accordance with the protocols approved by the local government authorities (Ministerium für Landwirtschaft, ländliche Räume, Europa und Verbraucherschutz, Kiel, Germany; AZ: 52-9/24, AZ: 85-8/19, AZ: 83-6/14; and by the Animal Care Committee of Santiago de Compostela University) and were performed in agreement with the Rules of Laboratory Animal Care and International Law on Animal Experimentation. Mice were kept at a

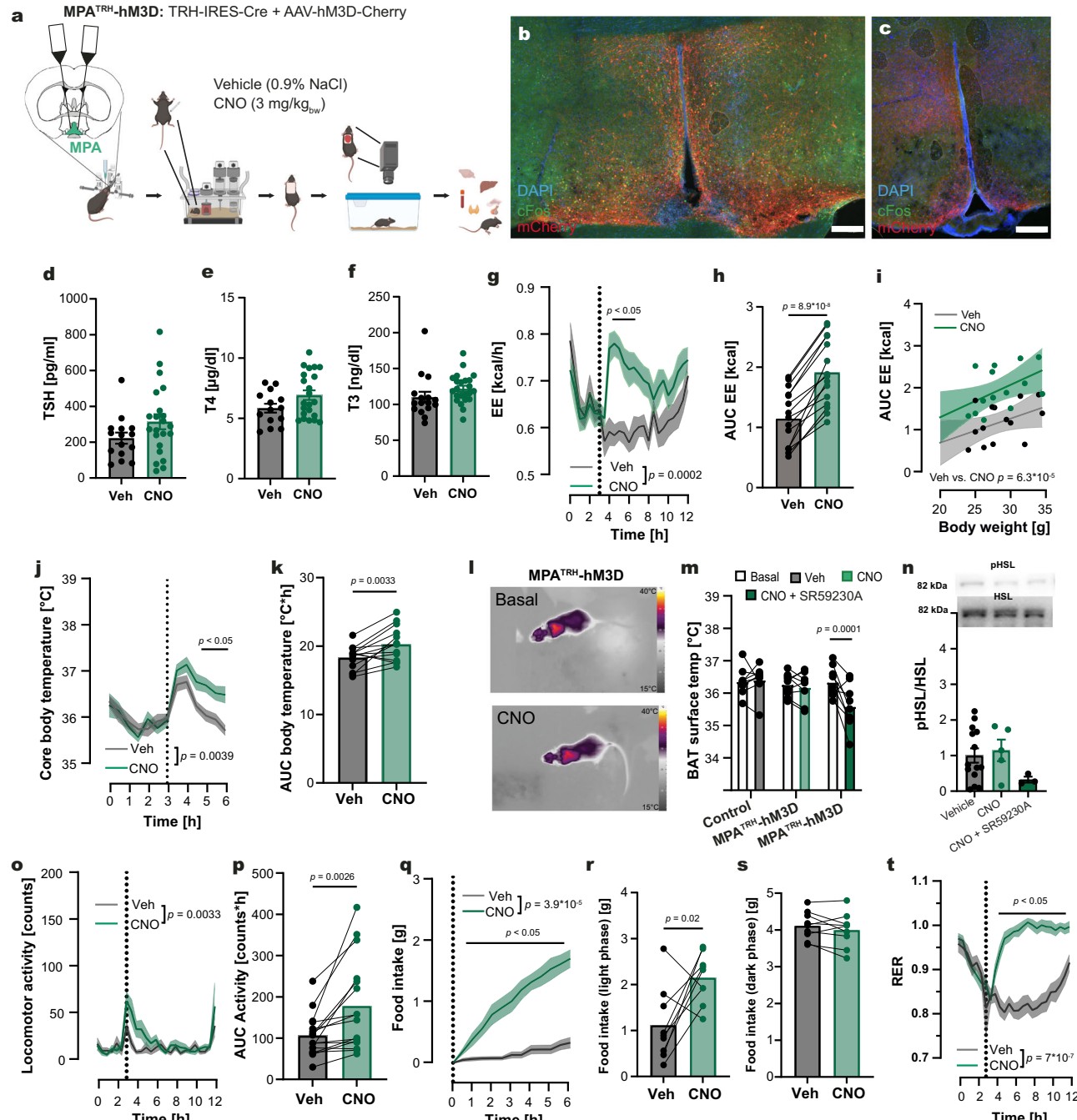

**Fig. 5 | Activation of TRH neurons in the MPA affects activity and energy homeostasis. a** Stereotactic injection of AAV-flex-hM3D-mCherry into the MPA of TRH-IRES-Cre mice generated MPA$^{TRH}$-hM3D mice. **b, c** Immunohistochemical validation of mCherry (red) and DREADD activation via cFos staining (green) 90 min after CNO injection in the MPA (**b**) and ME (**c**). Nuclei were counterstained with DAPI (blue). Scale bar: 100 μm. Staining was repeated in 10 animals. **d–f** Plasma levels of TSH (**d**) T4 (**e**), and T3 (**f**) measured 90 min after CNO (green) or vehicle (gray, Veh) treatment. **g, h** Energy expenditure (EE, **g**) and corresponding AUCs (**h**) after vehicle or CNO stimulation. **i** Linear regression of the changes of AUC$_{EE}$ from g against the body weight after vehicle (gray) and CNO (green) treatment. **j, k** Core body temperature (**j**) and as AUCs (**k**) following vehicle or CNO injection. **l** Representative IR-thermography images before and 45 min after CNO treatment. **m** Changes in BAT surface temperature before (white) and after treatment with vehicle (gray), CNO (green), or CNO + SR59230A (dark green). **n** Representative

Western blots of pHSL and total HSL in BAT 90 min after treatment with vehicle, CNO, or CNO + SR59230A (membranes in Supplementary Fig. 12). Quantification as ratio of pHSL/HSL. **o, p** Home cage locomotor activity (**o**) and corresponding AUCs (**p**) following CNO or vehicle treatment. **q–s** Food intake (**q**), and individual changes in total intake during the light (**r**) and dark (**s**) phases after CNO or vehicle treatment. **t** Respiratory exchange ratio (RER) curves during the light phase. Mice were treated in crossover design. Data are presented as mean ± s.e.m. For d-f, h, k, p, r, and s, two-tailed unpaired t-tests; for g, j, m, o, q, and t, two-way RM ANOVA followed by Šídák's multiple-comparisons test; for n, a one-way ANOVA test followed by Tukey's multiple comparisons test; For i, one-way analysis of covariance (ANCOVA) was used. Exact p-values, sample sizes and statistical details are provided in Supplementary Table 1. Source data are provided with this paper. Parts of figure a created in BioRender. *Schwaninger, M. (2026)* https://BioRender.com/xbrkg90.

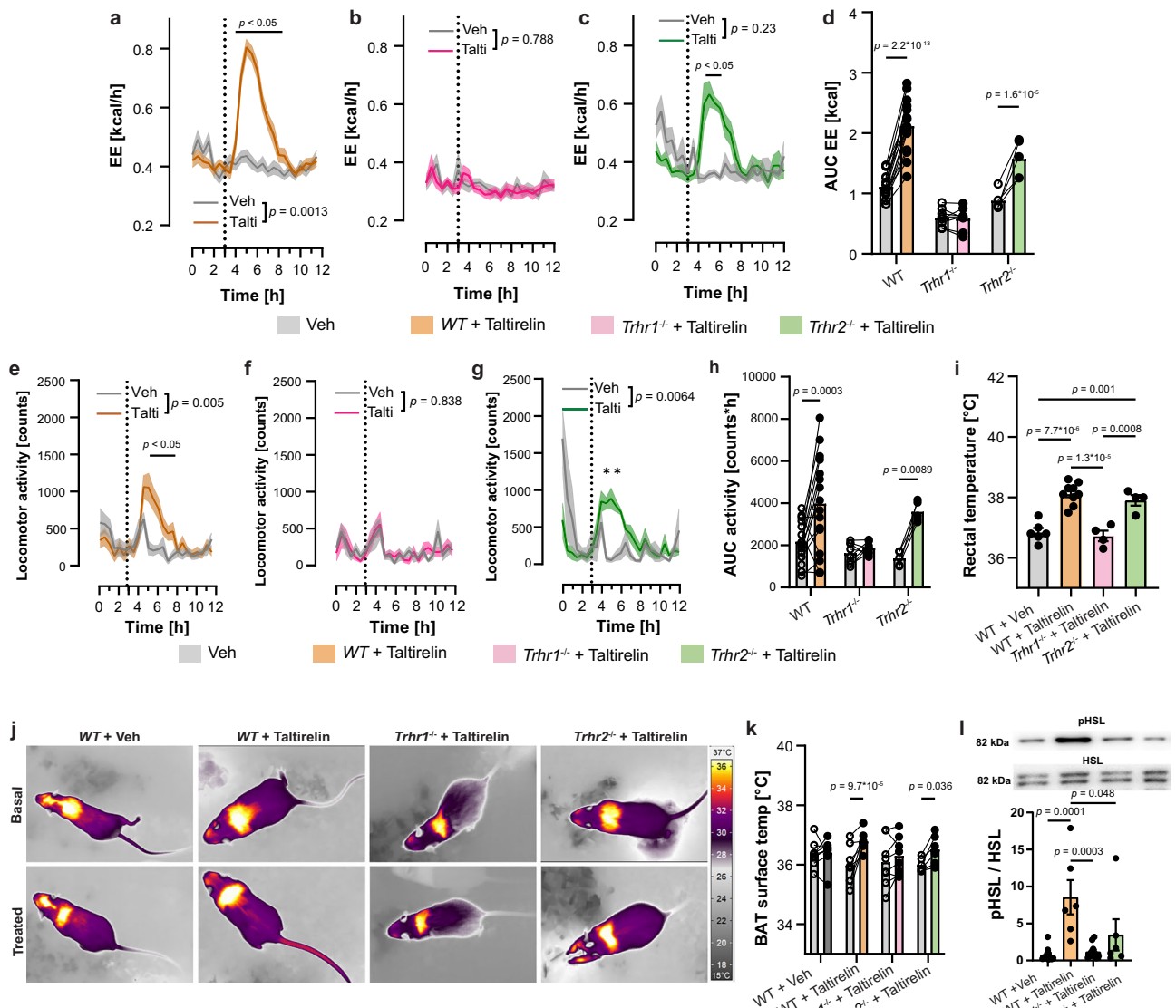

**Fig. 6 | Taltirelin increases energy metabolism via the TRHR1. a–c** Changes in energy expenditure (EE) after vehicle (0.9 % NaCl, gray) or taltirelin treatment (1 mg/kg$_{bw}$, i.p., colored lines) in *WT* (**a**, brown), *Trhr1$^{-/-}$* (**b**, pink), and *Trhr2$^{-/-}$* (**c**, green) mice. The dotted line indicates injection in a crossover design. **d)** Changes in EE between vehicle and taltirelin treatment are shown as AUC$_{EE}$ from curves shown in (**a–c**). **e–g** Changes in home cage locomotor activity for *WT* (**e**, brown), *Trhr1$^{-/-}$* (**f**, pink), and *Trhr2$^{-/-}$* (**g**, green) mice after taltirelin (colored line) and vehicle (gray) treatment. **h** Changes in home cage locomotor activity after the vehicle and taltirelin stimulation as AUC of activity curves from (**e–g**). **i** Rectal temperature in *WT* (brown), *Trhr1$^{-/-}$* (pink), and *Trhr2$^{-/-}$* (green) mice 90 min after taltirelin injection (1 mg/kg$_{bw}$; i.p.). Controls represented by vehicle injections in *WT* (gray, 0.9 % NaCl). **j** Exemplary infrared images of BAT surface temperature 5 min prior- (Basal) and 45 min post-injection of taltirelin or vehicle (Treated) in *WT*,

*Trhr1$^{-/-}$* and *Trhr2$^{-/-}$* mice. **k** Individual changes in BAT surface temperature before (Basal, light gray) and 45 min after treatment with taltirelin (colored bars) or vehicle (dark gray). **l** Representative western blots of pHSL and HSL of BAT 90 min after taltirelin or vehicle treatment (membranes see Supplementary Fig. 12). Changes in HSL phosphorylation in BAT are shown as the ratio of pHSL/HSL, for *WT* (brown), *Trhr1$^{-/-}$* (pink), and *Trhr2$^{-/-}$* (green) mice treated with taltirelin normalized to vehicle-treated *WT* mice (gray). Data presented as mean ± s.e.m. For (**d, e, h, k**), two-way RM ANOVA followed by Šídák's multiple-comparisons test; for (**a, b, c, f, g**), a REML mixed-effects model followed by Šídák's multiple-comparisons test; for i and l, one-way ANOVA followed by Šídák's multiple comparisons test was used. *: $p < 0.05$. Exact *p*-values, sample sizes and statistical details are provided in Supplementary Table 1. Source data are provided with this paper.

constant temperature (23 °C) on a 12 h light/dark cycle and were provided with a standard laboratory chow diet (2.98 kcal/g; Altromin, Hannover, Germany) and water *ad libitum*. Experiments were performed on adult mice of both sexes aged 8–20 weeks and were not analyzed for sex differences.

All mouse lines were established on a C57BL/6 N background. The mouse lines used were described previously: TRH-IRES-Cre$^{+/–}$[40], *Trhr1$^{-/-}$* [23,48], and *Trhr2$^{-/-}$* (obtained from Deltagen[48],). *Trhr1$^{-/-}$* and *Trhr2$^{-/-}$* were bred together to generate *Trhr1$^{-/-}$:Trhr2$^{-/-}$* double knockout mice. TRH-IRES-Cre$^{+/-}$ were bred with *Trhr1$^{-/-}$* to generate *Trhr1$^{-/-}$*TRH$^{Cre}$ mice. TRH-

IRES-Cre$^{+/-}$ were bred with *Trhr2$^{-/-}$* to generate *Trhr2$^{-/-}$* TRH$^{Cre}$ mice. TRH-IRES-Cre$^{+/-}$ were bred with *Trhr1$^{-/-}$:Trhr2$^{-/-}$* double knockout mice to generate *Trhr1$^{-/-}$:Trhr2$^{-/-}$*TRH$^{Cre}$ mice. Mice from the TRH-IRES-Cre line were used for the taltirelin experiments. For the site-specific expression of the activating hM3Dq-mCherry DREADD system in TRH neurons, we injected bilaterally the Cre-dependent AAV-CAG-flex(hM3D-mCherry) vector (2 × 10$^9$ gp; 75 nl) into the PVN, DMH, MPA or RPa of the mice. For the chronic inactivation of TRH neurons, we injected the Cre-dependent AAV-Syn1-flex(TeNT-2A-$^{nuc}$tdTomato) vector (2 × 10$^9$ gp; 75 nl) into the PVN, DMH, or MPA[44]. Experiments started at least

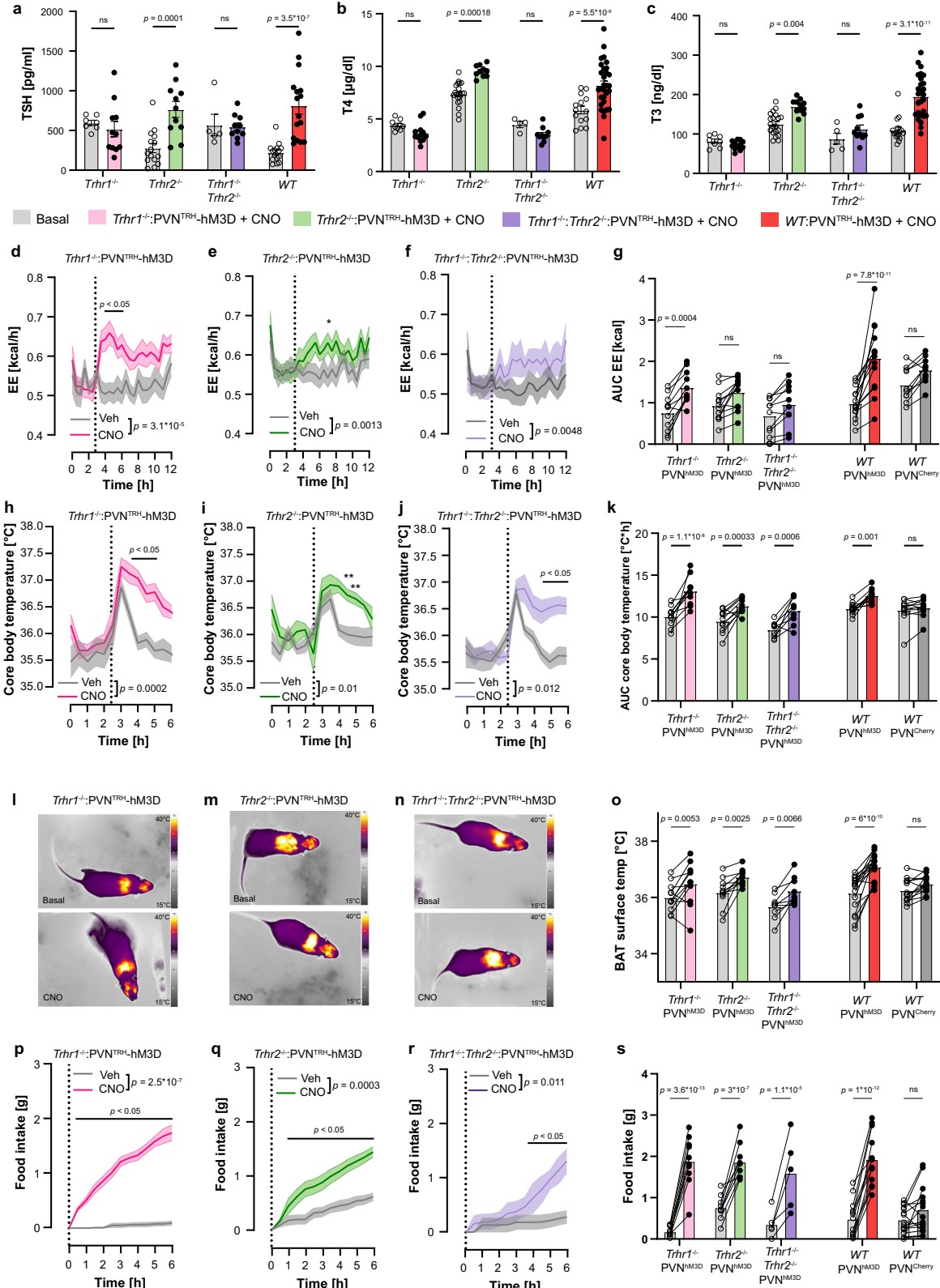

2 weeks after AAV injection to ensure proper expression of the transduced genes. Investigators were blinded to the treatment, genotype of mice, or both in all experiments and analysis. Calorimetric experiments were designed in a crossover experimental design, ensuring that each mouse received both the CNO (3 mg/kg body weight (bw)) and the vehicle (NaCl, 0.9 %). To test CNO-mediated effects, an additional control group of TRH-IRES-Cre$^{+/-}$ mice injected with AAV-CAG-

flex(mCherry) vector ($2 \times 10^9$ gp; 75 nl) into the regions was used alongside the individual vehicle control (TRH$^{Cherry}$).

After the injection of the AAVs, a temperature and activity telemetry transmitter was implanted intraperitoneally during the same surgery. Mice were anesthetized with ketamine ($65 \mu g/g_{bw}$, i.p.) and xylazine ($14 \mu g/g_{bw}$, i.p.). On a heating plate with rectal temperature measurement, the skin was disinfected (70 % ethanol), and an initial

**Fig. 7 | Contributions of TRH receptors in changes of energy homeostasis in PVN$^{TRH}$-hM3D mice. a–c** Changes of plasma levels of TSH (**a**) T4 (**b**) and T3 (**c**) in PVN$^{TRH}$-hM3D with *Trhr1* (pink, *Trhr1$^{-/-}$*:PVN$^{TRH}$-hM3D), *Trhr2* (green, *Trhr2$^{-/-}$*:PVN$^{TRH}$-hM3D), *Trhr1$^{-/-}$*:*Trhr2$^{-/-}$* double knockout (purple, *Trhr1$^{-/-}$*:*Trhr1$^{-/-}$*:PVN$^{TRH}$-hM3D) or *WT* (red, *WT*:PVN$^{TRH}$-hM3D) mice, 90 min after CNO treatment compared to basal levels of littermates with the same genotype (gray). **d–f** Changes in energy expenditure (EE) after CNO and vehicle treatment over time in *Trhr1$^{-/-}$*:PVN$^{TRH}$-hM3D (**d**, pink), *Trhr2$^{-/-}$*: PVN$^{TRH}$-hM3D (**e**, green) and *Trhr1$^{-/-}$*:*Trhr2$^{-/-}$*: PVN$^{TRH}$-hM3D (**f**, purple) mice. **g** Individual changes of EE between vehicle and CNO stimulation shown as AUC of EE curves from (**d–f**) as *WT*: PVN$^{TRH}$-hM3D (red) and *WT*:PVN$^{TRH}$-mCherry (dark gray). **h–j** Changes in core body temperature after CNO and vehicle treatment over time in *Trhr1$^{-/-}$*:PVN$^{TRH}$-hM3D (**h**, pink), *Trhr2$^{-/-}$*:PVN$^{TRH}$-hM3D (**i**, green), *Trhr1$^{-/-}$*:*Trhr2$^{-/-}$*:PVN$^{TRH}$-hM3D (**j**, purple) mice. **k** Individual changes of core body temperature between the vehicle (gray) and CNO stimulation are shown as AUC of core body temperature curves from (**h–j**) as *WT*:PVN$^{TRH}$-hM3D (red) and *WT*:PVN$^{TRH}$-mCherry (dark gray). **l–n** Exemplary IR-thermography pictures of *Trhr1$^{-/-}$*:PVN$^{TRH}$-hM3D (**l**), *Trhr2$^{-/-}$*:PVN$^{TRH}$-hM3D (**m**), *Trhr1$^{-/-}$*:*Trhr2$^{-/-}$*: PVN$^{TRH}$-hM3D (**n**) mice before (Basal) and 45 min after CNO injection. **o** Individual changes of BAT surface temperature between basal (gray) and CNO stimulated *Trhr1$^{-/-}$*: PVN$^{TRH}$-hM3D (pink), *Trhr2$^{-/-}$*: PVN$^{TRH}$-hM3D (green), *Trhr1$^{-/-}$*:*Trhr2$^{-/-}$*: PVN$^{TRH}$-hM3D (purple), *WT*: PVN$^{TRH}$-hM3D (red) and *WT*:PVN$^{TRH}$-mCherry (dark gray). **p–r** Changes in food intake after CNO and vehicle treatment over time in *Trhr1$^{-/-}$*:PVN$^{TRH}$-hM3D (**p**, pink), *Trhr2$^{-/-}$*:PVN$^{TRH}$-hM3D (**q**, green), *Trhr1$^{-/-}$*:*Trhr2$^{-/-}$*:PVN$^{TRH}$-hM3D (**r**, purple) mice. **s** Individual changes in food intake between vehicle (gray) and CNO stimulation shown as food intake over 6 h from (**p–r**) as *WT*:PVN$^{TRH}$-hM3D (red) and *WT*:PVN$^{TRH}$-mCherry (dark gray). All data presented as mean ± s.e.m. For (**a–h**) and (**j–s**), two-way RM ANOVA followed by Šídák's multiple-comparisons test; for i, a REML mixed-effects model followed by Šídák's multiple-comparisons test was used. *: $p < 0.05$; **: $p < 0.01$. Exact $p$-values, sample sizes and statistical details are provided in Supplementary Table 1. Source data are provided with this paper.

skin incision followed by a small abdominal incision was made. The sterile sensor (TA-F10; Data Science International) was placed in the abdominal cavity, and the peritoneum, as well as the skin were closed with absorbable sutures. At the end of the surgery, the isoflurane gas supply was stopped, and the animals were observed in their home cage on a heating plate (37 °C) until they were fully awake. After the operation, the mice were treated with carprofen (5 mg/kg BW; s.c., 5 µl/g$_{bw}$) every 24 h for 2 days. For radiotelemetric measurements of the core body temperature and activity, the cages were placed on a plate that continuously records the sensor data (DSI, Data Science International).

### Calorimetric measurements
Food intake, EE and home cage activity were measured using an open circuit indirect calorimetry system (PhenoMaster TSE Systems). Mice were previously single housed and acclimatized in training cages for 3 days before data acquisition to adapt them to the systems' food and water dispensers. On the experimental day, 2.5–3 h after the beginning of the light phase, mice were given an i.p. injection of vehicle and immediately returned to their cages. 24 h after the vehicle (0.9 % NaCl solution) treatment animals received the CNO (3 mg/kg$_{bw}$) injection and were measured for an additional 24 h cycle. Additionally, a separate cohort was investigated without access to food in the 6 h after injection of CNO.

### Cold tolerance test
Two weeks after transduction with the AAV-Syn1-flex(TeNT-2A-$^{nuc}$tdTomato), mice were single housed 4 days before the cold exposure and were acclimatized to the climatized cabinet at 23 °C for at least 18 h with the normal light/dark cycle. Three hours before the cold exposure, the cages were connected to a cage individual indirect calorimetry (CaloSys) to measure the EE each minute with water and food *ad libitum*. The core body temperature was measured every 2 min, by implanted telemetry sensors (TA-F10; Data Science International). At 12 noon, the ambient temperature was lowered to 10 °C, in 20 min, for 4 h. After the 4 h, the mice were immediately anesthetized with ketamine: 200 mg/kg$_{bw}$, xylazine: 24 mg/kg$_{bw}$; in 0.9 % saline solution, the body composition was measured (Bruker minispec Plus) and the animals were sacrificed.

### Retrograde tracing
For retrograde tracing, animals were first anesthetized using ketamine (65 µg/g$_{bw}$; i.p.) and xylazine (14 µg/g$_{bw}$; i.p.). Then the area of the iBAT was shaved and a skin incision was made. The tissue was carefully parted to reveal the iBAT pads on both sides. Pseudorabies-GFP tagged virus (Bartha strain) was loaded into a Hamilton syringe and 5 µl was delivered into each iBAT pad. The skin was then closed using vicryl suturing material. Animals were treated with carprofen

(5 mg/kg$_{bw}$; s.c.) post-surgery. Animals were sacrificed 5 days after surgery.

### IR thermography
The animals were shaved in the iBAT area with a long-hair trimmer under isoflurane (4 %) two days before the measurement. To measure the temperature of the iBAT, the lid of the cage was opened and the camera was positioned above the cage. Approximately 2–3 h after the start of the light phase, two baseline recordings were taken 30 min apart. After the baseline recordings, mice were injected intraperitoneally with NaCl (0.9 %), CNO (3 mg/kg$_{bw}$), taltirelin (1 mg/kg$_{bw}$), or CNO (3 mg/kg$_{bw}$) +SR59230A (10 mg/kg$_{bw}$), and 45 min after the treatment, a third recording was taken. Between injection and measurement, the mice had no access to food. The investigator was blinded to the substance. The camera recorded a video for 1 min with 60 frames per min. The videos were analyzed blinded, and the temperature of the iBAT was determined.

### Tissue collection
90 min after treatment with NaCl (0.9 %), CNO (3 mg/kg$_{bw}$), or CNO (3 mg/kg$_{bw}$) +SR59230A (10 mg/kg$_{bw}$) the animals were anesthetized with a mixture of ketamine and xylazine administered intraperitoneally (ketamine: 200 mg/kg$_{bw}$, xylazine: 24 mg/kg$_{bw}$; in 0.9 % saline solution). After achieving deep anesthesia, blood was first collected by puncturing the right ventricle, stabilized with EDTA (Microvette CB300), and centrifuged at 10,000 g for 5 min at 4 °C. Immediately after blood sampling, the animals were perfused with PBS. Brains and organs were quickly removed. The brains were postfixed in 4 % PFA overnight at 4 °C, then cryoprotected with sucrose solution (30 %, 4 °C, 24 h), and frozen in dry ice-cold 2-methylbutane. The other organs were snap-frozen in liquid nitrogen and stored at −80 °C.

### AAV production
rAAV with a mosaic capsid of the serotype 1 and 2 (1:1) were generated as described and purified by AVB Sepharose affinity chromatography. For each vector, the genomic titer was determined by quantitative PCR (qPCR) using primers against WPRE (WPRE forward primer: 5′-TGCCCGCTGCTGGAC-3′; WPRE reverse primer: 5′-CCGACAA-CACCACGGAATTG-3′) as described previously[48]. The AAV-Syn1-flex(-TeNT-2A-$^{nuc}$tdTomato) was ordered from Addgene as an AAV9 vector.

### Stereotaxic injections
For stereotaxic injections, mice were anesthetized with ketamine (65 µg/g$_{bw}$; i.p.) and xylazine (14 µg/g$_{bw}$; i.p.) plus carprofen (5 mg/kg$_{bw}$; s.c.) before they were fixed in the stereotaxic frame (David Kopf Instruments). Lidocaine (1 %; s.c.) was used as a local anesthetic in the skin above the skull. During the procedure, the body temperature of the animals was maintained at 37 °C using a heating plate. The

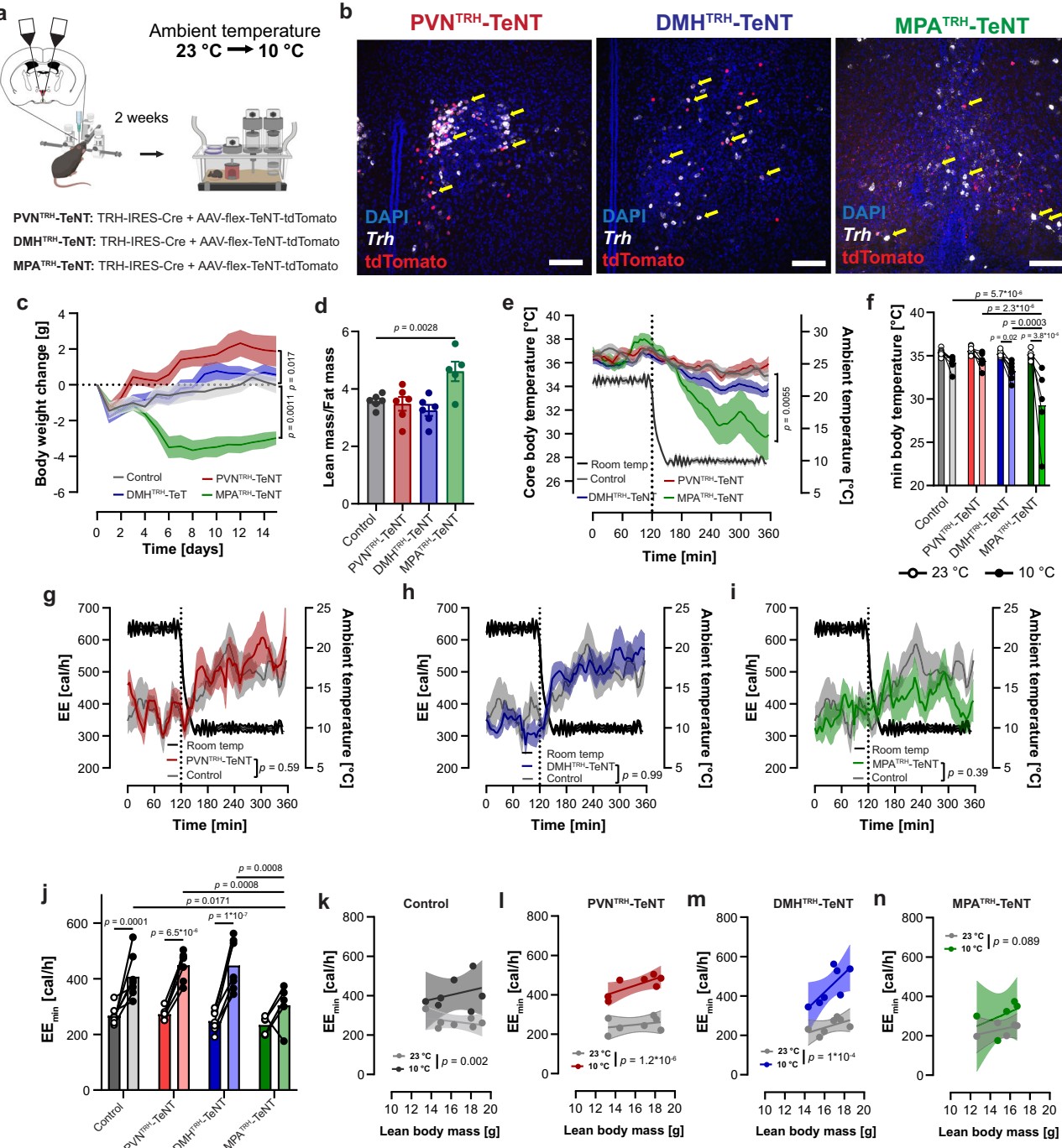

**Fig. 8 | Inhibition of TRH neurons in acute cold response. a** Stereotactic injection of AAV-flex-TeNT-Tomato in the PVN, DMH, and MPA of TRH-IRES-Cre mice led to the generation of PVN^TRH-TeNT, DMH^TRH-TeNT, and MPA^TRH-TeNT mice. AAV-flex-mCherry injected TRH-IRES-Cre mice were used as controls.
**b** Immunohistochemical validation by staining tdTomato (red) and *Trh* mRNA (white) by RNAScope. Nuclei stained by DAPI (blue). Scale bar: 100 μm. Staining was repeated in 6 animals/group. **c** Change in body weight after inactivation of TRH neurons. **d** Change in lean/fat mass ratio. **e** Change of core body temperature after changing the ambient temperature from 23 °C to 10 °C (black curve) in PVN^TRH-TeNT (red), DMH^TRH-TeNT (blue), MPA^TRH-TeNT (green) and control (gray) mice. **f** Comparison of minimum core body temperature at 23 °C (time 0-120 min) and after the change to 10 °C (time 240-360 min) ambient temperature in PVN^TRH-TeNT (red), DMH^TRH-TeNT (blue), MPA^TRH-TeNT (green) and control (gray) mice. **g–i** Change in energy expenditure (EE) after changing the ambient temperature from 23 °C to 10 °C (black curve) in PVN^TRH-TeNT (**g**, red), DMH^TRH-TeNT (**h**, blue),

MPA^TRH-TeNT (**i**, green) and control (**g-i**, gray) mice. **j** Comparison of minimum EE (EE_min) reached at 23 °C (time 0-120 min) and after the change to 10 °C (time 240-360 min) ambient temperature. **k–n** Linear regression of the individual changes of the EE_min from Fig. j) against the lean body mass at 23 °C (light gray) and 10 °C in control (**k**, gray), PVN^TRH-TeNT (**l**, red), DMH^TRH-TeNT (**m**, blue), and MPA^TRH-TeNT (**n**, green) mice. Data presented as mean ± s.e.m. For (**c, e, f, j**), two-way RM ANOVA followed by Šídák's multiple-comparisons test, for g-I, a REML mixed-effects model followed by Šídák's multiple-comparisons test; for (**d**), a one-way ANOVA followed by Šídák's multiple comparisons test; for (**k–n**), one-way analysis of covariance (ANCOVA) was used. Exact *p*-values, sample sizes and statistical details see Supplementary Table 1. Source data are provided with this paper. Traces for the individual animals are shown in Supplemental Figure 13. Source data are provided as a Source Data file. Parts of figure a created in BioRender. *Schwaninger, M. (2026)* https://BioRender.com/8tsy3rf.

following coordinates relative to the bregma were used to inject 75 nl of AAV into each hemisphere. PVN: AP = −0.75 mm, ML ± 0.3 mm, DV −5 mm; MPA: AP = 0.5 mm, ML = ± 0.2 mm, DV = −5 mm; DMH: AP = −1.9 mm, ML = ± 0.5 mm; DV = −5 mm; rostral RPa: AP = −5.3 mm, ML = ± 0 mm, DV = −5.75 mm. Injections were performed over 5 min and the capillary stayed in place for another 5 min to avoid backflow of the vectors. The scalp was sutured and the animals were returned to their home cage. They were treated with carprofen (5 mg/kg$_{bw}$; s.c.) for 2 days after surgery.

## Plasma hormone measurements
After treatment, plasma was collected to determine TSH, T4, and T3 concentrations. For TSH we used the MILLIPLEX MAP Mouse Pituitary Magnetic Bead Panel (MPTMAG-49K; Merk) and a Luminex system (Millipore) according to the manufacturer's instructions. Total T4 and T3 plasma concentrations were determined by ELISA (T3: NovaTec Immundiagnostica GmbH, DNOV053; T4: Drg Instruments GmbH, EIA-1781).

## In situ hybridization in mouse brains
The in situ hybridization was performed with the RNAScope® Multiplex Fluorescent Reagent Kit v2 (Advanced Cell Diagnostics, ACD, Hayward, CA, USA, used probes listed in Supplementary Table 4). We used a set of 20 double Z probes of the target genes. Briefly, mouse brains were extracted and post-fixed at 4 °C in freshly prepared 4 % PFA for 12 h, cryoprotected with sucrose solution (30%, 4 °C, 24 h), frozen in dry ice-cold 2-methylbutane and cut using a cryostat. Sections were stored at −20 °C in anti-freeze (30% ethylene glycol, 30% glycerol, 40% PBS) until staining. Sections (50 μm) were mounted on glass slides, dehydrated with ethanol, baked for 30 min at 37 °C, and treated with hydrogen peroxide and protein kinase IV. Then, the sections were incubated with Z probes for 2 h at 40 °C and hybridized sequentially using pre-amplifiers, amplifiers, and HRP-labeled oligonucleotides, followed by TSA probes labeled with different fluorophores. After in situ hybridization, the sections were further stained as described in the section on immunohistochemistry.

## Protocol for qPCR
Hypothalamic tissue of the ARC was dissected and washed with PBS and lysed using lysis buffer from the RNeasy® FFPE Kit (Qiagen). mRNA was isolated according to the manufacturer instructions. RNA was transcribed using Moloney Murine Leukemia Virus Reverse Transcriptase and random hexamer primers (Promega). Real-time PCR was performed with the FastStart Essential DNA Green Master (Roche) using a 3-Step amplification according to the following protocol: 10 s at 95 °C, 20 s at 60 °C and 20 s at 72 °C (45 cycles). The following primers were used for quantitative real-time PCR: Ppia forward: 5′- GCA TAC AGG TCC TGG CAT CT-3′, Ppia reverse: 5′- CAT CCA GCC ATT CAG TCT TGG-3′; Agrp forward: 5′- CGC TTC TTC AAT GCC TTT TGC-3′, Agrp reverse: 5′- ATT CTC ATC CCC TGC CTT TGC-3′; Npy forward: 5′- TGT GTT TGG GCA TTC TGG CT-3′, Npy reverse: 5′- GCT GGA TCT CTT GCC ATA TCT CT-3′; Pomc forward: 5′-AGC GTT ACG TGG CTT CA TGA-3′, Pomc reverse: 5′- TGG AAT GAG AAG ACC CCT GCA-3′, Quantified results were normalized to Ppia using the ΔΔCt method.

## Immunohistochemistry
Samples were cut at 30–50 μm thickness using a cryostat. Samples were either collected directly on glass slides or as free-floating sections in an anti-freeze solution and stored at −20 °C. For immunohistochemistry samples were washed 3 × 10 min with 0.3 % PBST (PBS + Triton-X 0.3 %) before blocking with 4% BSA in PBST 0.3% for 60 min. After blocking, primary antibodies (see Supplementary Tabl. 2) were diluted in blocking solution (1:500) and applied overnight at 4 °C. The next day, samples were washed 3 × 10 min with PBST 0.3 %. Secondary

antibodies (see Supplementary Tabl. 3) were diluted in blocking solution (1:1000) along with DAPI (1:1000). Samples were incubated with secondary antibody solution for 2 h at room temperature. Following this step, the samples were once more washed 3 × 10 min with PBST 0.3%. Free-floating slices were then transferred onto coated glass slides. All samples received a coverslip mounted with DAKO (Agilent, S302380-2) mounting medium. Images were taken by confocal laser scanning microscopes (Leica, SP5 or STELLARIS 5) or a fluorescence microscope (Leica, DMI 6000B). pictures were analyzed with Fiji (ImageJ 1.52i).

## DAB immunostaining against mCherry
The sections were washed three times for ten minutes each in PBS, then subjected to antigen retrieval by incubation at 85 °C for 30 min in 10 mM citrate buffer (pH 6.0). After cooling to RT, the sections were washed again in PBS. Subsequently, the sections were treated for 30 min at RT with 1% hydrogen peroxide in PBS, and after another washing step, they were blocked for one hour at RT in 10% BSA and 0.3% Triton X-100 in PBS. They were then incubated overnight at 4 °C with the primary mCherry antibody (1:4000, see Supplementary Table 2) in blocking solution. The next day, the sections were washed three times for 10 min in 0.1% Triton X-100 in PBS and then incubated for 2 h at RT with the biotinylated secondary antibody (1:250, anti-goat HRP) in 0.3% Triton X-100 in PBS. After washing, the sections were treated for one hour with the ABC solution from the Vectastain Elite® ABC-HRP Kit (Peroxidase, Standard; Vector Laboratories) and washed thoroughly again. The sections were then incubated with 3,3′-diaminobenzidine (DAB) substrate from the DAB Substrate Kit, Peroxidase HRP (Vector Laboratories). The staining reaction was monitored under a microscope (Leica, DMI 6000B) and stopped after one minute by transferring the sections into PBS. Following further washing in PBS, the sections were mounted onto slides in 0.02 M phosphate buffer, left to dry overnight, and on the following day dehydrated in 70, 96, and 100% ethanol followed by xylene, and finally coverslip using DPX mounting medium (Sigma Aldrich). Pictures were taken by a bright-field microscope (Leica, DMI 6000B). Location of cell bodies were located with Fiji (ImageJ 1.52i).

## Western blot
Protein lysates from the BAT were homogenized in lysis buffer (Cell Signaling Technology, #9803) with freshly added protease inhibitor phenylmethylsulfonyl fluoride (PMSF, Carl Roth 6367.2). The protein concentration was measured using Bradford Assay (Bradford reagent, Sigma-Aldrich B6916), and the total amount of protein was calculated for each tissue sample. The protein content for the samples was equalized when diluting the samples with SDS buffer (0.75 M Tris-HCl, 0.08 g/ml SDS, 40% glycerol, 0.4 mg/ml bromophenol blue, and 62 mg/ml DTT) and incubated at 95 °C for 10 min. The protein lysates (briefly incubated at 95 °C for 5 min before loading) were then subjected to SDS-PAGE and the proteins were transferred to polyvinylidene difluoride (PVDF, Bio-Rad 1620177) with a wet blotting system, blocked with 5% bovine serum albumin (BSA, Sigma Aldrich A1470), and incubated with different primary antibodies (pHSL [Ser660]; HSL, and alpha Tubulin, see Supplementary Table 2) diluted in blocking solution overnight at 4 °C. The membranes were then incubated in their respective HRP-conjugated secondary antibodies (either anti-rabbit or anti-mouse, see Supplementary Table 3), for 2 h at room temperature. The membranes were developed using enhanced chemiluminescence (SuperSignal West Pico Substrate, Thermo Scientific, 34580) and a digital detection system (Fusion Solo S, Viber). Immunoblots of pHSL, HSL, and tubulin were blinded and analyzed, using ImageJ (National Institutes of Health, RRID: SCR_002285). The intensity of the target protein was expressed relative to the intensity of tubulin and normalized to the ratio of the vehicle-treated control

group. To compare all membranes, an internal control (a pool of all samples) was loaded on each western blot. Each membrane was normalized to this internal control.

## Statistics

Data were analyzed using Prism 8 and 10 (GraphPad). Significance was considered when $p < 0.05$. Depending on the dataset and experimental design, different statistical methods were used as indicated in Supplementary Table 1. Parametric statistics (for example, $t$-test and ANOVA) were only applied if assumptions were met, that is, datasets were examined for Gaussian distribution using the D'Agostino–Pearson test, aided by visual inspection of the data and homogeneity of variances by Brown–Forsythe, Levene's or F-test (depending on the statistical method used). If assumptions for parametric procedures were not met or could not be reliably assumed due to a small sample size, non-parametric methods were used as indicated. Two-tailed tests were applied if not indicated otherwise. For comparing different treatments in the same animal, a repeated measurement (RM) analysis was used. In case of missing datapoints a mixed model was used. The mixed model assumed a compound symmetry covariance structure and was fitted using Restricted Maximum Likelihood (REML). Greenhouse-Geisser correction was used in ANOVA or mixed model (REML) statistics if the sphericity assumption was violated (Mauchly test). For analysis of the energy expenditure an ANCOVA was performed, using the body weight or lean body mass as a covariant (SPSS 28; IBM). Animals were randomly allocated to diet or treatment groups as long as age-matched, sex-matched, and littermate conditions were fulfilled. All analyses were performed blinded without the knowledge of the genotype or treatment if not needed for subsequent processing.

## Reporting summary

Further information on research design is available in the Nature Portfolio Reporting Summary linked to this article.

## Data availability

All data are available in the article and its Supplementary files or from the corresponding author upon request. Source data are provided with this paper.

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

## Acknowledgements

We would like to thank Bradford B. Lowell for kindly providing us with the TRH-IRES-Cre mice line. We are really grateful to Bryan Roth for giving us the original pAAV-hSyn-hM3D(Gq)-mCherry plasmid. We would also like to thank Pashkovski et al. for providing us the pAAV-hSyn-FLEX-TeLC-P2A-dTomato via Addgene. German Research Foundation (DFG): GRK1957 ("Adipocyte-Brain-Crosstalk") to H.M-F., J.W., and M.S.; DFG-Project-ID 424957847 - CRC/TRR296 "LocoTact" to J.M., H.H. and H.M-F.; DFG-Project-ID MU 3743/1-1 to H.M-F.; DFG-Project-ID 434396546 to R.O. European Research Council (ERC) Synergy Grant-2019-WATCH-810331 to M.S., R.N., and V.P.

## Author contributions

H.M-F. conceived the study; An.C., H.M-F., M.S., J.W. and R.O. designed the experiments; An.C., Ak.C., L.K., H.M-F., L.H, M.R., F.S., I.S., W.B., U.M., V.N. and N.D.A. performed and analyzed experiments; H.H., R.O., J.M., R.N., V.P. and M.S. provided essential tools and animal models. H.M-F., Ak.C and An.C. drafted the manuscript. An.C., H.M-F., M.S., J.W., R.O. Ak.C., L.K. L.H., M.R., F.S., I.S. W.B. U.M., V.N., N.D.A., H.H., J.M., R.N. and V.P. corrected the manuscript.

## Funding

## Competing interests

The Authors declare that they have no interests, financial resources, or employment relationships that improperly influence or affect the integrity of the submission.
