## [Transparent Peer Review file · Nature Communications]

Thyrotropin-releasing hormone neurons of different hypothalamic nuclei increase energy expenditure

Corresponding Author: Dr Helge Müller-Fielitz

Version 0:

Reviewer comments:

Reviewer #1

(Remarks to the Author)

Constantinescu et al. studied the role of TRH-expressing neurons in various brain regions in the regulation of energy balance. Through retrograde tracing, they found that multiple neuronal populations appear to provide polysynaptic input to the BAT. In addition, by chemogenetically activating neurons in different hypothalamic nuclei in TRH-Cre mice, the author found increases in BAT temperature, energy expenditure, and food intake. These effects were, in part, diminished in mice lacking *Trhr1* and/or *Trhr2*.

In overall, determining the circuits and functions of TRH-expressing cells beyond the neuroendocrine ones located in the paraventricular hypothalamus could be of interest. However, I have several concerns that need to be addressed:

1) While the authors demonstrate that chemogenetic stimulation of neurons in TRH-Cre mice has profound effects on feeding and energy expenditure, it remains unclear under which physiological conditions the artificially-activated neurons contribute to energy balance regulation. This is particularly the case for AAV expression in the preoptic area, the dorsomedial hypothalamus, and the raphe pallidus as the activity patterns and functions of *Trh*⁺ neurons in these areas have not been investigated. To provide this crucial information, the authors should monitor activity of the investigated neuronal populations in different conditions (e.g., cold exposure with Fos mapping), and, importantly, need to perform loss-of-function experiments.

2) Although it is likely that the targeted neurons for the chemogenetic activation studies express TRH, this has not been validated. Further, although the representative images suggest that specific targeting of neurons in hypothalamic nuclei was achieved, full validation of all included mice is missing. The latter is of particular consideration because rather large volumes (i.e., 75nl) were injected in rather small nuclei, which are in close proximity to each other (e.g., DMN and PVN). Extensive additional histological evaluation of all experiments is clearly needed, to confirm cellular and regional specificity.

3) Could it be that the increases in BAT and EE that the authors observed following artificial activation of neurons in the PVN and DMN, are driven indirectly by increases in food intake, at least in part? Performing experiments in absence of food should be performed, to exclude this critical confounding factor.

Reviewer #2

(Remarks to the Author)

This manuscript examined the metabolic functions of TRH neurons in specific brain regions: PVH, DMH, MPA and RPa. Chemogenetic activation of these neuronal populations in mice revealed diverse effects on energy homeostasis. PVH and DMH TRH neurons stimulated BAT activity through a multi-synaptic pathway, while MPA TRH neurons enhanced locomotor activity. Notably, these effects occurred independently of thyroid axis modulation, indicating that TRH neurons possess distinct, subtype-specific mechanisms for increasing energy expenditure beyond their known role in thyroid regulation.

This study presents an innovative concept and a methodologically robust investigation into the effects of TRH on energy balance, revealing its physiological relevance by demonstrating an independent action that challenges the traditional view that such effects are solely thyroid hormone (TH) dependent. Overall, I have a positive impression of the paper, as it significantly advances our understanding of central metabolic regulation. I do have a few minor recommendations:

1. Analysis of energy expenditure is convincing but the use of ANCOVA would also add significant value.
2. pHSL/HSL ratio provides a very interesting readout of BAT activation (in addition of BAT temperature), as indicative of lipolysis, given the quick observed effects. Did authors control the levels of UCP1 or other thermogenic markers? I assume the timing might be short for finding major changes in protein expression, but I am still curious about this.
3. The orexigenic effects of TRH manipulation are very interesting. Considering the interaction between TRH neurons in the (at least PVH TRH cells) and the ARC, analysis of AgRP and NPY expression would be of interest.
4. TRH is one of the first genes highly expressed in the hypothalamus of newborns. Traditionally, it was thought of as a mechanism to activate the HPT axis to induce thermogenesis. This study shows that TRH has direct effects (TH-independent). Some discussion would be of interest.

Reviewer #3

(Remarks to the Author)

Thyrotropin-releasing hormone neurons of different hypothalamic nuclei increase energy expenditure
Andreea Constantinescu... Helge Müller-Fielitz

Summary

Constantinescu and colleagues demonstrate the role of distinct TRH neuronal populations in stimulating energy expenditure through polysynaptic projections to the brown adipose tissue (BAT). Utilizing retrograde tracing, this study identifies numerous regions within the brain that are connected to the BAT, including TRH neurons in the PVN and DMH hypothalamic nuclei. Using chemogenetics and pharmacological approaches the present study demonstrates TRH neurons in the PVN stimulate BAT thermogenesis via the sympathetic nervous system. Similarly, activation of TRH neurons in the DMH increases energy expenditure and activates BAT via the sympathetic nervous system. Based on experiments using TRHR1/2 KO mice, the authors propose a major role for TRHR2 in mediating the PVN-TRH stimulation of energy expenditure. However, mice lacking both receptors retained the thermogenic response to PVN-TRH activation. Therefore, the authors suggest that this response is dependent on other neurotransmitter synthesized by TRH neurons.

Comments

This study presents a compelling and pertinent investigation into the multifaceted role of TRH populations in regulating thermogenesis and energy expenditure. By delving into the regulatory mechanisms of these populations, the study significantly contributes to the advancement of scientific knowledge within the field. While the role of TRH neurons in modulating neuroendocrine responses and motivated behaviors has been previously established, the characterization and regulation of specific TRH populations remain an area of ongoing research. Data from double knockouts (TRHR1/2) is intriguing and supports the notion that TRH is not required for mediating short-term regulation of brown adipose tissue (BAT) thermogenesis. Other researchers have made analogous observations concerning appetite regulation by distinct peptidergic neurons.

1. Alterations in food intake influence multiple endocrine systems, potentially impacting thermogenesis and energy expenditure. Given that TRH activation and taltirelin administration influence food intake, it becomes challenging to discern the primary effects from those secondary to food consumption. Conducting chemogenetics experiments in a food-deprived state would facilitate the analysis and discussion of these effects. Indeed, in the models used in Figures 2, 3 and 5 food intake was increased. Performing these experiments in a pair fed manner at least in the case of one model would be important.

2. Taltirelin's effects on EE and BAT thermogenesis are abolished in TRHR1KO mice but not in TRHR2KO mice. However, the effects of PVN-TRH activation on EE are dependent on TRHR2. This finding is interesting, and the manuscript would benefit from further elaboration on these results. Demonstrating where *Trhr2* is expressed and discussing these differences would be beneficial.

3. The authors have utilized activating chemogenetics to examine the role of various TRH neuronal populations. Surprisingly, they did not inhibit subsets of TRH neurons in context of stressors known to require increased EE ie cold. This could demonstrate the physiologic importance of TRH neurons.

4. The paragraph commencing at line 249 could be revised to enhance clarity. For instance, the sentence "Despite increased food intake" (253) follows a statement indicating that taltirelin reduced food intake during the dark phase. Does the 253 sentence refer to the light phase?

Version 1:

Reviewer comments:

Reviewer #1

(Remarks to the Author)

These authors have successfully addressed all of my concerns. I would like to congratulate them for this exciting work!

Reviewer #2

(Remarks to the Author)

All my comments have been addressed.

We thank the reviewers for the time they took to give us a thorough feedback. In response to the issues raised by the reviewers, we have now included new data which helped us improve the manuscript considerably. The changed text passages are marked in the main manuscript in red. Here, we would like to address the reviewer comments in a point-by-point manner.

Reviewer #1 (Remarks to the Author):

Constantinescu et al. studied the role of TRH-expressing neurons in various brain regions in the regulation of energy balance. Through retrograde tracing, they found that multiple neuronal populations appear to provide polysynaptic input to the BAT. In addition, by chemogenetically activating neurons in different hypothalamic nuclei in TRH-Cre mice, the author found increases in BAT temperature, energy expenditure, and food intake. These effects were, in part, diminished in mice lacking *Trhr1* and/or *Trhr2*. In overall, determining the circuits and functions of TRH-expressing cells beyond the neuroendocrine ones located in the paraventricular hypothalamus could be of interest.

Response: We are grateful for the overall positive evaluation.

However, I have several concerns that need to be addressed:

1) While the authors demonstrate that chemogenetic stimulation of neurons in TRH-Cre mice has profound effects on feeding and energy expenditure, it remains unclear under which physiological conditions the artificially-activated neurons contribute to energy balance regulation. This is particularly the case for AAV expression in the preoptic area, the dorsomedial hypothalamus, and the raphe pallidus as the activity patterns and functions of *Trh*⁺ neurons in these areas have not been investigated. To provide this crucial information, the authors should monitor activity of the investigated neuronal populations in different conditions (e.g., cold exposure with Fos mapping), and, importantly, need to perform loss-of-function experiments.

Response:

We thank the reviewer for this important question about the physiological function of the different TRH neuron populations.

To address this question, we performed cold exposure and assessed *Fos* mRNA expression using *in situ* hybridization (RNAScope) in the brain regions implicated in thermogenesis, as indicated by our data. In this experiment, mice were subjected to a 4-hour cold challenge and compared to control animals maintained at standard room temperature (23 °C). To specifically detect activation within *Trh*-expressing neurons, *Trh* mRNA was co-labeled in the same sections. Cold exposure resulted in increased *Fos* expression in TRH neurons within the PVN, MPA, and DMH. These results are included in the new version of the manuscript in **Supplementary Fig. 8a-i**.

Following the demonstration of cold-induced activation of TRH neurons, we performed a cold tolerance test in mice in which the TRH neurons had been chronically silenced via expression of the tetanus toxin light chain (TeNT). This was achieved by injecting AAV-hSyn1-flex-(TeNT-^{nu}tdTomato)¹ into the PVN (PVN^{TRH}-TeNT), DMH (DMH^{TRH}-TeNT), and MPA (MPA^{TRH}-TeNT) of TRH-IRES-Cre mice (**Fig. 8a**). Co-expression of the nuclear tdTomato allowed us to confirm effective targeting and expression in the relevant brain regions (**Fig. 8b**). Expression of TeNT has already been shown to silence TRH neurons in the DMH². After viral transduction, animals were characterized under baseline conditions for changes in body weight (**Fig. 8c**), body composition (**Fig. 8d**, and **Supplementary Fig. 8j-l**) and food intake (**Supplementary Fig. 8m-n**). Consistent with previous findings², DMH^{TRH}-TeNT mice showed a shift in food intake from the

dark phase to the light phase, resulting in a redistribution of feeding across the circadian cycle without altering total daily food intake (**Supplementary Fig. 8m-n**). These data highlight a role for DMH TRH neurons in the circadian regulation of feeding behavior and confirmed the data of Douglass *et al* ². Silencing of TRH neurons in the PVN and MPA led to alterations in body weight (**Fig. 8c**). PVN^{TRH}-TeNT mice exhibited a modest increase in body weight, while MPA^{TRH}-TeNT mice showed a dramatic weight loss. The reduction in MPA^{TRH}-TeNT mice began approximately five days after TeNT expression and progressed such that two animals had to be removed from the experiment for ethical reasons (**Fig. 8c**). The body weight loss in MPA^{TRH}-TeNT mice was accompanied by an increase in the lean-to-fat mass ratio (**Fig. 8d**), indicating a substantial metabolic impact. Fifteen days after neuronal silencing, mice were subjected to a cold tolerance test. Upon cold exposure, core body temperature dropped significantly in both DMH^{TRH}-TeNT and MPA^{TRH}-TeNT mice, compared to the control mice (**Fig. 8e, f**), indicating an impaired cold tolerance. In MPA^{TRH}-TeNT mice, this was further accompanied by a marked reduction in energy expenditure under cold conditions (**Fig. 8g-n**). The individual traces for EE, activity and core body temperature for each animal are included in **Supplementary Fig. 13**.

These results clearly demonstrate a role for DMH^{TRH} neurons in the regulation of feeding behavior and cold tolerance, while MPA^{TRH} neurons appear to be essential for both body weight regulation and cold tolerance. The full dataset is presented in **Fig. 8**, **Supplementary Fig. 8**, and **Supplementary Fig. 13**. The results are included in the text in **Lines 325-353**.

2) Although it is likely that the targeted neurons for the chemogenetic activation studies express TRH, this has not been validated. Further, although the representative images suggest that specific targeting of neurons in hypothalamic nuclei was achieved, full validation of all included mice is missing. The latter is of particular consideration because rather large volumes (i.e., 75nl) were injected in rather small nuclei, which are in close proximity to each other (e.g., DMN and PVN). Extensive additional histological evaluation of all experiments is clearly needed, to confirm cellular and regional specificity.

Response:

We would like to thank the reviewer for this question. Before performing the experiments, coordinates and injection parameters were optimized. The same AAV batch was used throughout the entire series, including the animals used in the revision that were subjected to food deprivation experiments. To confirm the specificity of the DREADD system expression in TRH neurons, we performed additional *in situ* hybridization of *Trh* and *Fos*, together with immunostaining for mCherry as a marker for the hM3D expression by the AAV transduction. The mCherry expression was confined to the target nuclei. We now show this additional data in **Supplementary Figure 2a-b**, **Supplementary Figure 3a-b** and **Supplementary Figure 5a-b**. In the MPA, DMH, and PVN, we further show that the mCherry-positive cells are positive for *Trh* mRNA and exhibit an increase in *Fos* mRNA expression **Supplementary Fig. 2b-c**, **3b-c**, and **5b-c**. This confirms the ability of our method to specifically target TRH neurons in specific nuclei and in turn activate them as indicated by the large proportion of *Trh*-positive neurons expressing *Fos*. The corresponding images, including those of the new animals investigated during the revision, are now presented in **Supplementary Fig. 2, 3, and 5**.

As requested, to validate the injection sites and a possible spread of the AAV into other nuclei, we now show the distribution of mCherry-positive cell bodies in MPA, PVN and DMH of 12 representative animals from different cohorts. No significant spread to the other brain areas was detected, which allowed us to further validate our injection parameters. This distribution is shown and documented in the maps in the

new **Supplementary Fig. 11**. We would like to emphasize that these stainings were performed in each animal and for every brain section to confirm the specificity of the injection. Only animals in which we targeted the TRH neurons in the nuclei of interest were used for the analysis.

3) Could it be that the increases in BAT and EE that the authors observed following artificial activation of neurons in the PVN and DMN, are driven indirectly by increases in food intake, at least in part? Performing experiments in absence of food should be performed, to exclude this critical confounding factor.

Response:

We thank the reviewer for the important question about an indirect effect of food intake on the energy expenditure and temperature regulation.

We apologize that the description of the BAT thermography measurements was incomplete. The increase in BAT temperature was recorded 45 min after CNO injection just before sacrificing the animals. In this period the animals had no access to food. This excludes any effect of food intake on the increase in BAT temperature after activation of TRH neurons. We have now adjusted the methods section accordingly (**see Line 577**).

To investigate the influence of TRH neurons on EE and core body temperature, independent of food intake, we activated the MPA^{TRH}, PVN^{TRH}, and DMH^{TRH} neurons by CNO injection and removed the food parallelly. The mice were deprived of food for the following 6 h after CNO injection. The mice were injected with CNO and NaCl, as controls, on the two days before the food deprivation experiment. The increase in EE after CNO stimulation remained stable after food deprivation (shown for PVN^{TRH}-hM3D: **Supplementary Fig. 2f-h, Lines 181-185**; DMH^{TRH}-hM3D: **Supplementary Fig. 3f-h, Lines 209-211**; and MPA^{TRH}-hM3D: **Supplementary Fig. 5f-h, Lines 240-242**). In the PVN^{TRH}-hM3D and DMH^{TRH}-hM3D the body temperature was also increased under food deprivation, thus confirming, additionally, the activation of BAT in these two groups. Interestingly, approximately 2 to 3 hours after stimulation with CNO, both PVN^{TRH}-hM3D and MPA^{TRH}-hM3D mice experienced a sharp drop in body temperature, which suggests an imbalance in the energy homeostasis in these mice without access to food. We have included these important data into the new **Supplementary Fig. 2** (for PVN^{TRH}-hM3D), **Supplementary Fig. 3** (for DMH^{TRH}-hM3D), and **Supplementary Fig. 5** (for MPA^{TRH}-hM3D).

Reviewer #2 (Remarks to the Author):

This manuscript examined the metabolic functions of TRH neurons in specific brain regions: PVH, DMH, MPA and RPa. Chemogenetic activation of these neuronal populations in mice revealed diverse effects on energy homeostasis. PVH and DMH TRH neurons stimulated BAT activity through a multi-synaptic pathway, while MPA TRH neurons enhanced locomotor activity. Notably, these effects occurred independently of thyroid axis modulation, indicating that TRH neurons possess distinct, subtype-specific mechanisms for increasing energy expenditure beyond their known role in thyroid regulation.

This study presents an innovative concept and a methodologically robust investigation into the effects of TRH on energy balance, revealing its physiological relevance by demonstrating an independent action that challenges the traditional view that such effects are solely thyroid hormone (TH) dependent. Overall, I

have a positive impression of the paper, as it significantly advances our understanding of central metabolic regulation.

Response: We thank the reviewer for the kind words about our manuscript

I do have a few minor recommendations:

1. Analysis of energy expenditure is convincing but the use of ANCOVA would also add significant value.

Response:

We thank the reviewer for the comment regarding the analysis of energy expenditure. We have now reanalyzed all the energy expenditure data. As suggested by the reviewer, we performed an ANCOVA using body weight as a covariate. This analysis demonstrated that the observed changes following the activation of different TRH subpopulations remain consistent even when body weight is included as a covariate. We have replaced the body weight normalized evaluations of energy expenditure with the unnormalized values and included the corresponding ANCOVA analyses in the relevant sections and Figures. In addition, the body weight normalized energy expenditure values are shown in the respective Supplementary Figures.

2. pHSL/HSL ratio provides a very interesting readout of BAT activation (in addition of BAT temperature), as indicative of lipolysis, given the quick observed effects. Did authors control the levels of UCP1 or other thermogenic markers? I assume the timing might be short for finding major changes in protein expression, but I am still curious about this.

Response:

As noted by the reviewer, and as demonstrated by others, UCP1 expression is typically upregulated approximately 4-6 hours³ after stimulation. We performed western blot analyses on our samples and observed an increase in UCP1 protein in just PVN^{TRH}-hM3D mice (Letter Figure 1), which would, in principle, support our hypothesis. Given the short period after activation of the neurons, this finding was unexpected, and we consider it unlikely to reflect a robust physiological regulation. Instead, it could represent a time-dependent limitation inherent to our experimental design. Due to this possibility and to avoid potential misinterpretation of a rapid UCP1 protein regulation by TRH neurons, we have decided not to include these results in the manuscript.

Letter Figure 1

Changes in the UCP1/tubulin protein ratio, determined by Western blot analysis of brown adipose tissue, for vehicle- (gray), CNO- (red, blue, and green), or CNO + SR59230A-treated (dark red, dark blue, and dark green) PVN^{TRH}-hM3D (red), DMH^{TRH}-hM3D (blue), and MPA^{TRH}-hM3D (green) mice.

3. The orexigenic effects of TRH manipulation are very interesting. Considering the interaction between TRH neurons in the (at least PVH TRH cells) and the ARC, analysis of AgRP and NPY expression would be of interest.

Response:

To address the reviewer's comment, we examined *Fos* mRNA expression in the arcuate nucleus of animals with altered food intake after stimulation with CNO. We found that, compared to mCherry-injected control mice, *Fos* was upregulated in the neurons of the arcuate nucleus in PVN^{TRH}-hM3D, DMH^{TRH}-hM3D and MPA^{TRH}-hM3D animals. To a large extent, *Fos*-positive cells could be superimposed on AgRP-positive cells. In contrast to *Fos*, the expression of *Agrp*, *Npy*, and *Pomc* mRNA was not altered in the animals. These data are presented in the new **Supplementary Fig. 9, Lines 174-177, Line 209, and Line 252.**

4. TRH is one of the first genes highly expressed in the hypothalamus of newborns. Traditionally, it was thought of as a mechanism to activate the HPT axis to induce thermogenesis. This study shows that TRH has direct effects (TH-independent). Some discussion would be of interest.

We thank the reviewer for the insightful comment regarding the possible involvement of the hypothalamic TRH neurons in the activation of brown adipose tissue (BAT) in newborns. We have addressed this point and incorporated it into the discussion section of the manuscript (**Lines 417-429**).

Reviewer #3 (Remarks to the Author):

Thyrotropin-releasing hormone neurons of different hypothalamic nuclei increase energy expenditure
Andreea Constantinescu... Helge Müller-Fielitz

Summary

Constantinescu and colleagues demonstrate the role of distinct TRH neuronal populations in stimulating energy expenditure through polysynaptic projections to the brown adipose tissue (BAT). Utilizing retrograde tracing, this study identifies numerous regions within the brain that are connected to the BAT, including TRH neurons in the PVN and DMH hypothalamic nuclei. Using chemogenetics and pharmacological approaches the present study demonstrates TRH neurons in the PVN stimulate BAT thermogenesis via the sympathetic nervous system. Similarly, activation of TRH neurons in the DMH increases energy expenditure and activates BAT via the sympathetic nervous system. Based on experiments using TRHR1/2 KO mice, the authors propose a major role for TRHR2 in mediating the PVN-TRH stimulation of energy expenditure. However, mice lacking both receptors retained the thermogenic response to PVN-TRH activation. Therefore, the authors suggest that this response is dependent on other neurotransmitter synthesized by TRH neurons.

Comments

This study presents a compelling and pertinent investigation into the multifaceted role of TRH populations

in regulating thermogenesis and energy expenditure. By delving into the regulatory mechanisms of these populations, the study significantly contributes to the advancement of scientific knowledge within the field. While the role of TRH neurons in modulating neuroendocrine responses and motivated behaviors has been previously established, the characterization and regulation of specific TRH populations remain an area of ongoing research. Data from double knockouts (TRHR1/2) is intriguing and supports the notion that TRH is not required for mediating short-term regulation of brown adipose tissue (BAT) thermogenesis. Other researchers have made analogous observations concerning appetite regulation by distinct peptidergic neurons.

We thank the reviewer for the positive assessment of our manuscript.

1. Alterations in food intake influence multiple endocrine systems, potentially impacting thermogenesis and energy expenditure. Given that TRH activation and taltirelin administration influence food intake, it becomes challenging to discern the primary effects from those secondary to food consumption. Conducting chemogenetics experiments in a food-deprived state would facilitate the analysis and discussion of these effects. Indeed, in the models used in Figures 2, 3 and 5 food intake was increased. Performing these experiments in a pair fed manner at least in the case of one model would be important.

Response:

We thank the reviewer for this comment. As suggested by the reviewer, we have now performed chemogenetics experiments under food-deprived conditions to be able to differentiate the effects directly mediated by food intake from the effects of activation of TRH neurons in the different nuclei. We designed the experiments such that the same mice underwent CNO and NaCl treatments on consecutive days followed by CNO paralleled with a 6 h fasting period. The EE increase observed during TRH neuron activation remained stable despite food deprivation, for all three populations stimulated (shown for PVN^{TRH}-hM3D: **Supplementary Fig. 2f-h**; DMH^{TRH}-hM3D: **Supplementary Fig. 3f-h**; and MPA^{TRH}-hM3D: **Supplementary Fig. 5f-h**). This proves that the increase in EE is mediated by the activation of TRH neurons and is not due to the food intake following the activation of these neurons. Additionally, we can also show that the increase in body temperature following the activation of PVN^{TRH} and DMH^{TRH} persisted despite food deprivation, which confirms the role of these two TRH neuron populations in activating the BAT. Interestingly, approximately 2 to 3 hours after CNO stimulation, PVN^{TRH}-hM3D and MPA^{TRH}-hM3D mice experienced a sharp drop in body temperature, illustrating an imbalance in the energy homeostasis in these mice when denied access to food. These findings suggest that food intake induced by TRH neuron activation in the PVN and MPA is essential for maintaining energy homeostasis. Particularly noteworthy is the central role of MPA-TRH neurons, which appear to be critically involved in body weight regulation. We have included these important data into the new **Supplementary Figure 2** (for PVN^{TRH}-hM3D), **Supplementary Figure 3** (for DMH^{TRH}-hM3D), and **Supplementary Figure 5** (for MPA^{TRH}-hM3D).

With these results we are able to identify two specific phases following activation of TRH neurons under food restriction. The first phase shows an increase in EE and body temperature, which are both independent of food intake. However, food intake plays an important role, starting approximately 3 h after activation of TRH neurons, in the maintenance of energy homeostasis and body temperature. Furthermore, we thank the reviewer for the well-considered suggestion of a pair feed experiment. We see some limiting factors for performing pair feed experiments. We measured the food intake after TRH neuron activation during the light phase, when mice consume less food in general, which helps us detect an effect of increase in food intake and EE. Mice consume between 0.5 -1 g food during the light phase.

We expect, that in a pair feed experimental setup, the control animals would consume the whole portion of food, when made available, thus confounding food intake and EE measurements. This would make it difficult to interpret the effects of TRH neuron activation. Therefore, we decided to perform the food-deprivation experiment, which confirm a food-independent regulation in the energy homeostasis of the TRH neurons.

2. Taltirelin's effects on EE and BAT thermogenesis are abolished in TRHR1KO mice but not in TRHR2KO mice. However, the effects of PVN-TRH activation on EE are dependent on TRHR2. This finding is interesting, and the manuscript would benefit from further elaboration on these results. Demonstrating where *Trhr2* is expressed and discussing these differences would be beneficial.

Response:

We thank the reviewer for this valuable question. To address it, we examined the projection areas of PVN^{TRH} neurons and performed RNAScope *in situ* hybridization to assess expression patterns of *Trhr1* and *Trhr2* mRNA. Following CNO injection in PVN^{TRH}-hM3D mice, we observed, an increase in *Fos* expression in areas where we identified the projections, as shown in the new **Supplementary Fig. 10**. The increase in *Fos* expression was particularly evident in the lateral septum, the thalamus, and the mammillary nucleus. In parallel, we found that *Trhr1* and *Trhr2* mRNA were differentially expressed in these areas, often with distinct spatial patterns. Specifically, *Trhr1* mRNA was strongly expressed in the lateral septum and in the medial part of the mammillary nucleus adjacent to the third ventricle. In contrast, *Trhr2* mRNA expression was more prominent in the striatum, thalamus, the lateral part of the mammillary nucleus, and parts of the caudal lateral hypothalamus, which is consistent with the projectome data of Jiao *et al* 2025 ⁴. Importantly, we were able to colocalize the *Trhr2* mRNA with *Fos* in CNO treated PVN^{TRH}-hM3D mice. All three *Trhr2* containing regions have previously been implicated in the regulation of energy homeostasis ^{5,6}. Whether they are critical mediators of the observed effects remains to be determined in future studies. We have added these findings to the Results and Discussion sections of the revised manuscript (**Supplementary Figure 10, Lines 146-149, Lines 303-306**).

3. The authors have utilized activating chemogenetics to examine the role of various TRH neuronal populations. Surprisingly, they did not inhibit subsets of TRH neurons in context of stressors known to require increased EE ie cold. This could demonstrate the physiologic importance of TRH neurons.

Response:

We thank the reviewer for raising this important point regarding the role of TRH neurons in thermoregulation. To address this topic, we first assessed cold-induced activation of TRH neurons. Mice were exposed to a 4-hour cold challenge, and *Fos* mRNA expression was analyzed in *Trh*-expressing neurons using RNAScope *in situ* hybridization. We observed a significant activation of TRH neurons in the PVN, DMH, and MPA of cold-exposed animals compared to controls at room temperature (**Supplementary Fig. 8a-i**).

To evaluate the functional relevance of the hypothalamic TRH neurons, we chronically silenced TRH-expressing neurons in MPA, PVN, and DMH via AAV-mediated expression of the tetanus toxin light chain (TeNT) in TRH-IRES-Cre mice using the AAV-hSyn1-flex-(TeNT-^{nucl}tdTomato)¹. Successful targeting was confirmed by tdTomato co-expression with *Trh* mRNA (**Fig. 8a-b**). Under baseline conditions, silencing of DMH^{TRH} neurons led to a redistribution of food intake across the circadian cycle without affecting total food intake (**Supplementary Fig. 8m-n**). This finding is consistent with the recently published data of

Douglass *et al*² which confirmed the functionality of this approach. Strikingly, silencing of MPA^{TRH} neurons resulted in a progressive and severe loss of body weight, accompanied by a reduction in lean/fat mass ratio (**Fig. 8c, d**, and **Supplementary Fig. 8j-l**), demonstrating a critical role of the MPA^{TRH} neurons in metabolic regulation. Fifteen days post-silencing, a cold tolerance test revealed that both DMH^{TRH}-TeNT and MPA^{TRH}-TeNT mice were unable to maintain core body temperature under cold exposure (**Fig. 8e, f**). In MPA^{TRH}-silenced animals, this was accompanied by a marked reduction in EE (**Fig. 8g-n**), indicating impaired thermogenesis. The individual traces for EE, activity and core body temperature for each animal are included in **Supplementary Figure 13**.

Together, these results demonstrate that TRH neurons in the DMH and MPA are involved in cold adaptation and energy balance, with MPA^{TRH} neurons playing a particularly important role in maintaining body weight and thermogenic capacity. The full dataset is presented in **Figure 8, Supplementary Figure 8, Supplementary Figure 13 and Lines 325-353**.

4. The paragraph commencing at line 249 could be revised to enhance clarity. For instance, the sentence “Despite increased food intake” (253) follows a statement indicating that taltirelin reduced food intake during the dark phase. Does the 253 sentence refer to the light phase?

Response: Thank you for pointing out this lack of clarity. We have revised the respective paragraph to present the results more clearly and comprehensively (**Lines 271-276**).

1. Pashkovski, S.L., *et al*. Structure and flexibility in cortical representations of odour space. *Nature* **583**, 253-258 (2020).
2. Douglass, A.M., *et al*. Acute and circadian feedforward regulation of agouti-related peptide hunger neurons. *Cell metabolism* (2024).
3. Nedergaard, J. & Cannon, B. UCP1 mRNA does not produce heat. *Biochim Biophys Acta* **1831**, 943-949 (2013).
4. Jiao, Z., *et al*. Projectome-based characterization of hypothalamic peptidergic neurons in male mice. *Nat Neurosci* **28**, 1073-1088 (2025).
5. Zhang, J., Chen, D., Sweeney, P. & Yang, Y. An excitatory ventromedial hypothalamus to paraventricular thalamus circuit that suppresses food intake. *Nat Commun* **11**, 6326 (2020).
6. Sanchez-Jaramillo, E., *et al*. Origin of thyrotropin-releasing hormone neurons that innervate the tuberomammillary nuclei. *Brain structure & function* **227**, 2329-2347 (2022).